# Reliable Classifications with Guaranteed Confidence using the Dempster-Shafer Theory of Evidence

## Abstract

Reliably capturing predictive uncertainty is indispensable for the deployment of machine learning (ML) models in safety-critical domains. The most commonly used approaches to uncertainty quantification are, however, either computationally costly in inference or incapable of capturing different types of uncertainty (i.e., aleatoric and epistemic). In this paper, we tackle this issue using the Dempster-Shafer theory of evidence, which only recently gained attention as a tool to estimate uncertainty in ML. By training a neural network to return a generalized probability measure and combining it with conformal prediction, we obtain set predictions with guaranteed user-specified confidence. We test our method on various datasets and empirically show that it reflects uncertainty more reliably than a calibrated classifier with softmax output, since our approach yields smaller and hence more informative prediction sets at the same bounded error level in particular for samples with high epistemic uncertainty. In order to deal with the exponential scaling inherent to classifiers within Dempster-Shafer theory, we introduce a second approach with reduced complexity, which also returns smaller sets than the comparative method, even on large classification tasks with more than 40 distinct labels. Our results indicate that the proposed methods are promising approaches to obtain reliable and informative predictions in the presence of both aleatoric and epistemic uncertainty in only one forward-pass through the network.

## 1 Introduction

Both scientific and industrial challenges are increasingly addressed using machine learning (ML) methods. There is a growing tendency to employ ML even in scenarios where false predictions can have severe consequences, such as in medical diagnostics or autonomous driving. A major challenge for the development of ML functions in these kinds of applications is the reliability of the underlying model. Besides robustness of the ML module, having an estimate of the predictive uncertainty is an essential component.

While common metrics such as accuracy indicate how good a model is *on average*, they do not provide information about how certain it is for a *specific* input example. For instance, a pedestrian in traffic most likely doesn't mind that the autonomous driving module correctly classifies 99% of pedestrians, but fails to classify them and causes an accident. Needless to say, it becomes inevitable to have a measure of uncertainty for each instance that allows for making decisions that control instance-sensitive risk functions. In the instance of the pedestrian, this would, e.g., imply braking whenever the uncertainty of the class "pedestrian" is high.

In general, a distinction is drawn between two different sources of uncertainty, namely aleatoric and epistemic (Hüllermeier & Waegeman, 2021). Aleatoric, sometimes also referred to as statistical uncertainty, is associated with an inherent random nature of the information. This includes, for instance, noisy or imprecise data. Epistemic uncertainty, in contrast, stems from a lack of information (ignorance) of the decision maker about the perfect model, which can occur in the ML context due to a non-optimal training or because of an ill-chosen hypothesis space that does not include the perfect model. More data, i.e., information can reduce parts of epistemic uncertainty, but not aleatoric uncertainty, which is why the latter is sometimes also referred to as irreducible uncertainty. Reliable

ML methods ought to be capable of distinguishing between aleatoric and epistemic of uncertainty for some applications. For example, in active learning the samples of high epistemic uncertainty are leveraged to reduce uncertainty in regions of the feature space in which the model lacks sufficient data (Aggarwal et al., 2014).

There are applications, such as autonomous driving, which require fast inference times, as latency in the vehicle should be low. At the same time, such low latency applications may need to run on embedded devices, where computational resources are limited. Many common methods for quantifying aleatoric and epistemic uncertainty require sampling steps or utilize ensembles. For example, Bayesian neural networks sample from the posterior during inference, and an ensemble ML model can easily estimate uncertainty as deviation of predictions between individual ensemble members but multiplies the computational cost by the size of the ensemble. Thus, neither of these methods is suitable to be employed in a low latency setting with limited computational resources. In conclusion, a method that is simultaneously able to estimate aleatoric and epistemic uncertainty while being computationally lean is so far lacking and forms the starting point of our investigations.

**Contributions.** In this work, we introduce a novel approach to quantify predictive uncertainty that combines a neural network classifier based on the Dempster-Shafer theory of evidence (DST) (Dempster, 1967a;b; Shafer, 1976) with conformal prediction (CP) (Vovk et al., 2005; Papadopoulos et al., 2002; Lei & Wasserman, 2014), yielding reliable set predictions with guaranteed confidence for classification tasks subject to uncertainty in only one forward-pass through the network. Our approach can be adopted to arbitrary network architectures, only requiring to adjust the output layer to match the dimension of the quantities from DST. We construct a loss function that enables our classifier to be informative under different levels of epistemic uncertainty and we show empirically on different datasets that our method returns smaller, i.e., more informative sets than a comparable standard classifier network using a softmax output. Further, we introduce a reduced approach limited to false negative control that does not require an adaptation of the output dimension while still yielding informative set predictions.

## 2 RELATED WORK

In the field of uncertainty quantification in ML, various techniques have emerged to enhance the reliability and robustness of predictive models. This section reviews the state of the art in uncertainty quantification for single-label classification tasks (i.e., where each instance has only one correct label), discussing methods such as Bayesian neural networks (BNNs) (MacKay, 1992) or ensemble methods (Lakshminarayanan et al., 2017) and highlights the recent use of the Dempster-Shafer theory in Deep Learning.

A naïve strategy for capturing predictive uncertainty is the probabilistic approach, which involves a prediction of the discrete probability distribution over the different outcomes. Depicted in a probability simplex, this would constitute a point prediction, see Fig. 1(a). While being a reasonable starting point, there are indications that standard probabilistic models are often poorly calibrated (Guo et al., 2017), meaning that their output frequently does not accurately represent the true likelihood of the outcomes. To overcome this issue there exist several calibration methods, among which temperature scaling of the softmax parameter, introduced in Guo et al. (2017), and its extensions (Ji et al., 2019; Yu et al., 2022; Joy et al., 2023) is one of the most prominent post-processing techniques. Moreover, probabilistic approaches come with the drawback of not being able to distinguish between aleatoric and epistemic uncertainty. For certain applications, however, a distinction would be essential, especially when decisions can be rejected or delayed (Chow, 1970), or in active learning scenarios (Aggarwal et al., 2014).

More sophisticated techniques, such as BNNs or ensemble methods like Monte Carlo dropout (Gal & Ghahramani, 2016) or Bagging techniques (Breiman, 1996), are commonly used to capture a more accurate estimation of uncertainty in ML models and also allow to draw a distinction between aleatoric and epistemic uncertainty. At the heart of these methods is a distribution over possible sets of neural network parameters which is sampled multiple times for each data point during inference, cf. Fig. 1(b)-(c). The epistemic uncertainty is reflected by the variance of this distribution, while the aleatoric uncertainty is defined by the mean. Although this allows a distinction of the sources of uncertainty, those methods are computationally expensive in inference and therefore not suited for some tasks such as autonomous driving.

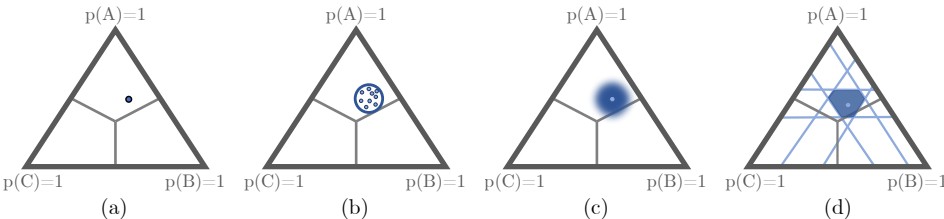

Figure 1: Probability simplex of three mutually exclusive events, denoted as $A$, $B$ and $C$ for different methods to capture uncertainty: While probabilistic point predictions only reflect aleatoric uncertainty (a), more advanced approaches like ensemble methods provide a measure of epistemic uncertainty in terms of the width of the scatter of the point predictions of multiple ensemble members (b). Similarly, in Bayesian neural networks, the variance of the distribution on the weights captures epistemic uncertainty (c). Yet other approaches, such as DST, provide credal sets whose size represents a measure of epistemic uncertainty (d).

Besides the expression of predictive uncertainty via distributions, there is the possibility of a representation via sets (Grycko, 1993). In our work, we make use of such set-valued predictions, i.e., the model outputs a *set* of possible labels, with the size of the set being an indicator of its certainty. Within this research area there are again two parallel branches. The first one is a pure post-processing method referred to as conformal prediction (Vovk et al., 2005; Shafer & Vovk, 2008; Balasubramanian et al., 2014), where sets with guaranteed confidence levels are returned. Since we use this method in our work, we formally introduce it in Section 3.2. In the second branch, sets are returned directly from the model. A special case of this is the so-called classification with reject option, where a classifier can express its uncertainty by refusing to classify a specific example (Herbei & Wegkamp, 2006). Other approaches, including our work, predict generalized probabilities for all sets of outcomes, which can be used to construct a credal set, cf. Fig. 1(d), defined as a convex set of probability distributions over all outcomes (see also Section 3.1 below), or utilize credal sets as representation of epistemic uncertainty, e.g., (Lienen & Hüllermeier, 2021).

To the best of our knowledge, only a few recent works directly make use of Dempster-Shafer theory to obtain set-valued predictions for quantifying predictive uncertainty. There also exists a rich body on belief functions (Cuzzolin, 2014), a connection between belief functions and conformal prediction has recently been found (Cella & Martin, 2022). Directly constructing outputs according to DST, Manchingal et al. (2023) learn feature vectors to fit Gaussian mixture models (GMMs), aiming to obtain a predetermined number of relevant subsets. The ability of GMMs to represent epistemic uncertainty, however, can be questioned, since they represent data density rather than prediction uncertainty.

The most directly related work to the approach presented in this paper is that in Manchingal & Cuzzolin (2022). They propose a neural network that assigns probability values based on the framework of DST to each set of outcomes (i.e., exponentially many values in the number of outcomes) by zero-padding label vectors and standard training. However, as they conclude in their work, this approach is similar to a traditional output in terms of training and validation. Our approach differs from that of Manchingal et al. in that we derive a novel loss function motivated by how uncertainty is represented in DST. Additionally, by modifying the output layer of a common neural network classifier, we are able to solve the scaling problem while still obtaining informative set predictions.

## 3 Preliminaries

The method developed in this work for quantifying predictive uncertainties is based on the Dempster-Shafer theory of evidence, which has only recently made its way into the uncertainty quantification community (Manchingal & Cuzzolin, 2022). Since the principles of DST are not common, they will be explained for completeness in Section 3.1 below. We combine a classifier based on DST with conformal prediction as a post-processing step, which we will elaborate on in Section 3.2.

### 3.1 DEMPSTER-SHAFER THEORY OF EVIDENCE

Dempster-Shafer theory (DST), sometimes also referred to as the theory of evidence or Dempster-Shafer theory of evidence, is a mathematical framework for decision-making in situations where evidence or information is incomplete, imprecise, or conflicting. The fundamental idea is to assign probabilities not to individual outcomes but to sets of outcomes (Dempster, 1967a), which addresses the limitation of standard Bayesian probability theory: expressing a state of ignorance, i.e., reflecting what you *don't* know without the need of committing to a prior. Consider for instance tossing a coin which is known to be fair. One would then assign both outcomes, $heads$ and $tails$, an equal probability of 0.5. If however, the coin is *not* known to be fair (and you don't have any further evidence about it's state), Bayesian probability theory would still assign a 0.5 probability to both outcomes, whereas DST allows to express the lack of knowledge by assigning a generalized probability (which is called *mass* function in DST) of 1 to the total set $\{heads, tails\}$ without the attempt to ascribe probabilities to the options *fair* and *not fair*.

**Definition 1** *Consider a so-called frame of discernment $\Theta = \{X_1, X_2, ..., X_n\}$, which is a set of $n$ mutually exclusive outcomes of the system under consideration. A basic probability assignment over $\Theta$, hereafter just denoted as mass function $m$, is a function that assigns a probability to each element in the powerset of $\Theta$, i.e., $m : 2^\Theta \to [0, 1]$ such that the two conditions hold:*

$$m(\emptyset) = 0 \tag{1}$$

$$\sum_{A \subseteq \Theta} m(A) = 1. \tag{2}$$

That is, the mass function $m(A)$ is an expression of how much one believes in *precisely* the set $A$ and not in any subset of $A$. To make further notation easier, we call $\mathbf{m} = (m(\emptyset), m(\{1\}), \ldots, m(\Theta)) \in \mathbb{R}^{2^n}$ the mass vector. Based on the mass one can derive two other functions over subsets of $\Theta$, namely belief and plausibility (Shafer, 1990).

**Definition 2** *Let $m$ be a mass function over $\Theta$. The belief function $bel$ and plausibility function $pl$ defined on the elements of the powerset $2^\Theta$ induced by $m$ are defined as*

$$bel(A) = \sum_{B \subseteq A} m(B) \tag{3}$$

$$pl(A) = \sum_{B \cap A \neq \emptyset} m(B) = 1 - bel(\bar{A}) \quad \forall A \subseteq \Theta \tag{4}$$

Belief and plausibility fulfill the property $bel(A) \leq pl(A)$ for all $A \subseteq \Theta$, which is why they are sometimes interpreted as lower and upper probability, respectively, for the particular set of outcomes. Similarly as for the mass function, a plausibility and a belief vector can be defined as $\mathbf{pl} = (pl(\emptyset), pl(\{1\}), \ldots, pl(\Theta))$ and $\mathbf{bel} = (bel(\emptyset), bel(\{1\}), \ldots, bel(\Theta))$. Note that neither $\mathbf{pl}$ nor $\mathbf{bel}$ are necessarily normalized. The plausibility and belief values of the singleton sets in $2^\Theta$ define a so-called credal set, which is a convex set of probability distributions over elementary events. The volume of such a convex set can serve as a measure of epistemic uncertainty, cf. Fig. 1(d). That is, once one knows the entire mass vector, an estimate of the epistemic uncertainty can be obtained. Recently, it has been mathematically proven that the volume of a credal set is a reliable measure of epistemic uncertainty only in the binary case (Sale et al., 2023), since the volume is zero in higher dimensions if the credal set is not of full dimensionality (in DST this occurs if plausibility and belief of a singleton are equal). Restricting the considerations to credal sets fully defined by their belief and plausibility bounds, alternative measures of epistemic uncertainty could be defined, which avoid the collapse issue in higher dimensions. For a single-label classifier, e.g., the mean belief-plausibility gap across all singleton sets could be used as a meaningful measure. Note that the presented axiomatic formulation of the Dempster-Shafer theory is the approach in Shafer (2008), which is not the only way to roll out DST. In their milestone work (Dempster, 1967a), Arthur P. Dempster defined mass, belief, and plausibility functions via a multi-valued map. Other routes include compatibility relations (Shafer, 1987) and random subsets (Nguyen, 1978). We refer the interested reader to Shafer (1990), where an overview over different formalizations is given. For completeness, we also highlight some criticisms (Zadeh, 1979; Pearl, 1988b;a) and their possible resolutions (Haenni, 2005; Wilson, 1992; Yager, 1987; Smets, 1992; Fixsen & Mahler, 1997), but we do not utilize these results.

## 3.2 Conformal Prediction

Conformal prediction (CP) provides a general methodology for prediction tasks aimed to obtain instance-wise set predictions with a guaranteed user-specified confidence (Vovk et al., 2005; Papadopoulos et al., 2002; Lei & Wasserman, 2014). It can be employed as a post-hoc calibration step to modify black-box point prediction schemes to produce reliable uncertainty estimates.

Consider a model $\hat{f}$ fitted to the problem under consideration, which outputs a prediction $\hat{y}$ from the output space $\mathcal{Y}$ given an input $x$ from the input space $\mathcal{X}$, i.e., $\hat{y} = \hat{f}(x)$ for true outcome $y$. Conformal prediction relies on the prescription of a scoring function $s$ (often also referred to as a non-conformity measure), which quantifies the extent that the prediction $\hat{y}$ deviates from the true output $y$. A common choice of a scoring function for a classifier that outputs softmax scores for each class ($\hat{f}(x) \in [0,1]^k$ for $k$ different outcomes) is one minus the softmax output of the true class, i.e., $s(x, y) = 1 - \hat{f}(x)_y$. It is further assumed that there is access to a held-back calibration dataset $\mathcal{D} = \{z_i = (x_i, y_i)\}_{i=1}^n$ of size $n$. The objective is then to predict a reliable output for a new test sample $z_{n+1} = (x_{n+1}, y_{n+1})$, of which the true output is unknown . For this purpose, the following assumption is made.

**Assumption 1** *Calibration dataset $\mathcal{D}$ and test sample $z$ are finitely exchangeable random variables. That is, there exists a joint probability distribution $p(\mathcal{D}, z)$ which is invariant under any permutation $\pi$ of $\{1, 2, ..., n + 1\}$, i.e.,*

$$p(z_1, z_2, ..., z_{n+1}) = p(z_{\pi(1)}, z_{\pi(2)}, ..., z_{\pi(n+1)}). \tag{5}$$

Note that the standard assumption in machine learning of independent and identically distributed (i.i.d.) random variables satisfies assumption 1.
In order to obtain a set predictor $\tau(x|\mathcal{D})$ given a calibration dataset $\mathcal{D}$ and a user-specified confidence level $1 - \alpha$, where $\alpha \in (0, 1)$ (e.g., $1 - \alpha = 0.9$) the following three steps need to be made.

1. Compute non-conformity measures for all samples in the calibration set $\mathcal{D}$, resulting in a set of scores $\{s(z_i)\}_{i=1}^n$.
2. Compute the $(1 - \alpha)$-quantile $Q_{1-\alpha}(\{s(z_i)\}_{i=1}^n)$ on the set of scores.
3. For a new sample $z_{n+1} = (x_{n+1}, y_{n+1})$ obtain the set predictor $\tau(x_{n+1})$ by including all candidate outputs $y' \in \mathcal{Y}$ whose scores $s(x_{n+1}, y')$ fall below the $(1 - \alpha)$-quantile, i.e.,

$$\tau(x_{n+1}|\mathcal{D}) = \{y' \in \mathcal{Y} : s(x_{n+1}, y') \leq Q_{1-\alpha}(\{s(z_i)\}_{i=1}^n)\} \tag{6}$$

By following these steps and under Assumption 1, CP guarantees that the set predictor $\tau(x_{n+1}|\mathcal{D})$ is calibrated, leading to Theorem 1.

**Theorem 1** *Under Assumption 1, for any confidence level $\alpha \in (0, 1)$ and for any scoring function $s$, the set predictor in (6) is well-calibrated, i.e., the probability that the true outcome $y$ for a given $x$ is contained in the set is larger or equal to $1 - \alpha$*

$$p(y \in \tau(x|\mathcal{D})) \geq 1 - \alpha. \tag{7}$$

A proof of Theorem 1 can be found in Vovk et al. (2005). Since the trivial set $\tau(x|\mathcal{D}) = \mathcal{Y}$ is always valid (in the sense that it always contains the true label), the returned set size can be understood as a measure of informativeness of the underlying model $\hat{f}$, also sometimes denoted as *sharpness* (Wang et al., 2023). The average set size obtained via CP can therefore be utilized as a quality measure, i.e., an evaluation metric for set predictors.

In addition to the here presented generic approach, several extensions to CP exist in literature, e.g., Bates et al. (2021) where the authors tackle CP for classification problems in which some mistakes are more severe than others, or the work of Mortier et al. (2021) where methods of utility maximization and CP are combined to balance between the correctness of the prediction and the set size. In the context of conformal prediction under ambiguous ground truth, it was recently demonstrated in Stutz et al. (2023) that it is necessary to take uncertainty into account during calibration to obtain valid prediction sets, and they introduce a new calibration procedure that accounts for this uncertainty.

## 4 METHOD

In this section, we develop a method based on DST to capture aleatoric and epistemic uncertainty for classification tasks in ML contexts. Note that recent theoretical results show how a second order model, i.e., a model capturing a distribution of probabilities, is necessary to represent epistemic uncertainty (Bengs et al., 2022) and how a standard training using loss functions without additional assumptions cannot learn epistemic uncertainty (Bengs et al., 2023). By using DST, we fulfill the requirement to utilize a second order model. Using an implicit assumption on the presence of epistemic uncertainty for the loss function design enables us to move beyond the latter restriction.

We consider the following learning problem. Given are pairs of i.i.d. samples $(x, y) \in \mathcal{X} \times \mathcal{Y}$ from a joint probability distribution $p_{\mathcal{X} \times \mathcal{Y}}$. We denote with $x \in \mathcal{X}$ an instance from the input space and $y \in \mathcal{Y}$ is called a label from the label space. Further, let $\mathcal{Y}$ be discrete with $n$ different labels, i.e., $\mathcal{Y} = \{1, \ldots, n\}$. Our method promotes a basic classifier of any kind into a probabilistic set predictor $\hat{h} : \mathcal{X} \to 2^{\mathcal{Y}}$ that outputs a mass vector from DST, cf. Fig. 2. The function $\hat{h}$ is expected to have the property that it assigns higher mass to larger sets for instances $x$ with high epistemic uncertainty. In cases of high aleatoric uncertainty, $\hat{h}$ should distribute the mass equally among the respective singleton sets related to the uncertainty. For low predictive uncertainty, the correct singleton set for the label $y$ belonging to the input $x$ is supposed to have high mass.

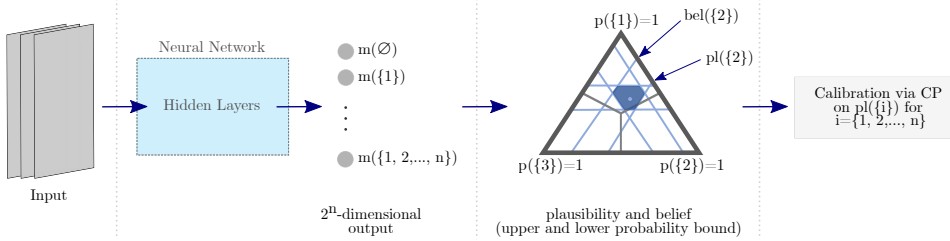

Figure 2: Overview of the general methodology of the DS Classifier. See Fig. 8 in Appendix C for a comparison of both proposed methods. We employ a neural network with arbitrary architecture except for the output layer and interpret the output of the model as mass vector. After training on a suitable loss function, e.g., (8), plausibilities of singleton sets are determined from the mass vectors for all samples in the calibration set. A calibration according to CP is performed on the plausibilities of the singleton sets of the true labels subsequently.

The difficulty in finding such a model for realistic use cases is that the true uncertainty is in general unknown, let alone the difficulty of being able to distinguish between aleatoric and epistemic uncertainty. Therefore, we introduce a specific loss function $\mathcal{L}_m$ that uses a trade-off hyperparameter $\lambda \in [0, 1]$ which, for small (large) values, makes the model tend to put high masses on larger (smaller) sets. Implicitly, this assumes that one is able to estimate how much of the uncertainty should be attributed to aleatoric and epistemic uncertainty during training. While being a strong assumption, we find it relatively unrestrictive in practice and observe improvements without the need to fine-tune $\lambda$. The loss then takes the form

$$\mathcal{L}_m = (1 - \lambda) \mathcal{L}_{\text{MSE}_0}(\hat{\mathbf{pl}}, \mathbf{pl}) + \lambda \mathcal{L}_{\text{MSE}}(\hat{\mathbf{bel}}, \mathbf{bel}). \tag{8}$$

Here, $\hat{\mathbf{pl}}$ and $\hat{\mathbf{bel}}$ are the plausibility and belief outputs, respectively, that result from the output mass of the model $\hat{h}$ and are constructed according to eq. (4) and (3). The vectors $\mathbf{pl}$ and $\mathbf{bel}$ are plausibility and belief vector resulting from the mass vector which has the entire mass on the singleton set defined by the label $y$, i.e., $m(\{y\}) = 1$. Note that the mean squared error (MSE) on the plausibility values has a subscript of 0. By this we indicate a small modification of the standard MSE that replaces all negative entries in the difference $\mathbf{pl} - \hat{\mathbf{pl}}$ by zero, in order to push more mass on the larger sets if $\lambda$ is small. We further want to emphasize that the choice of the loss function is not unique, but the one presented here was found to perform well empirically. In Appendix D we introduce an alternative training strategy.

In particular, it should be mentioned that our method, which we denote as *Dempster-Shafer (DS) Classifier* from now on, is applicable to arbitrary network architectures if the dimension of the

output layer is adjusted to match the dimension resulting from DST. The exponential scaling of the output layer ultimately limits the applicability of our method to small classification problems. This scaling has previously been addressed by the use of focal sets (Manchingal et al., 2023), which deliberately constrains the frame of discernment (see Def. 1) to the most relevant parts.

Here, we propose a restriction useful for the common case of false negative control by interpreting the outputs of the model as the plausibilities of the singleton sets directly and hence do not need to construct them in a subsequent processing step. The only adjustment to standard models in our second approach is that outputs are not normalized to 1. Instead, we apply a sigmoid function to the output (as opposed to a conventional softmax function). Similarly, for this method, we have constructed a loss function $\mathcal{L}_{\text{pl}}$ with hyperparameter $\lambda$, see eq. (9), that has similar properties as the loss function in (8) (small $\lambda$ pushes to high plausibilities in all outcomes, large lambda to a high plausibility only for the true outcome).

$$\mathcal{L}_{\text{pl}} = (1 - \lambda)\,\mathcal{L}_{\text{CE}}(\hat{\mathbf{pl}}_{\text{sgl}}, \mathbf{pl}_{\text{sgl}}) + \lambda\,\mathcal{L}_{\text{MSE}}(\hat{\mathbf{pl}}_{\text{sgl}}, \mathbf{pl}_{\text{sgl}}). \tag{9}$$

We use $\hat{\mathbf{pl}}_{\text{sgl}}$ to denote the predicted singleton plausibilities, and $\mathbf{pl}_{\text{sgl}}$ to indicate the one-hot encoded plausibilities, which have a one in the position of the true label and are zero otherwise. Intuitively, the cross entropy loss (CE) pushes more weight to all outcomes as it does not penalize to put more weight to the plausibility of the wrong outcome (they are multiplied by zero due to the one-hot encoding of the label). That is, for $\lambda = 0$ the loss function is minimized by all outcomes that predict a plausibility of 1 on the correct singleton set, regardless of the other predicted plausibilities. In particular, the state of maximum uncertainty, i.e., plausibility of 1 on all singletons, minimizes $\mathcal{L}_{\text{pl}}$ for $\lambda = 0$. Increasing the $\lambda$-parameter in favor of the MSE will lead to a more focused prediction with high plausibility only on one singleton set. We refer to this second approach as *DS Classifier (n-dimensional)* in the following.

To retrieve set predictions with guaranteed confidence from the DST quantities, we further apply CP to the the plausibilities of the singleton sets of the true label of the respective sample for both proposed approaches (note that we could do the calibration equivalently on one minus plausibility to match the description in Sec. 3.2). For the standard DS Classifier this implies computing the plausibilities from the mass vector. By calibrating on the plausibilities we achieve a false negative (FN) control, as in the inference stage only those labels are included in the prediction set with a plausibility exceeding the previously determined quantile. Note that while we are only utilizing plausibilities as upper bounds on the probability for FN control here, credal sets are more information rich and may prove useful for controlling diverse risk functions. It should be emphasized that in our second method we lose the ability to distinguish between aleatoric and epistemic uncertainty, although both types can still be implicitly captured by the same modeling assumptions. Correspondingly, we observe similar behavior for false negative control in the numerical experiments in Section 5 for both the n-dimensional and standard DS Classifier.

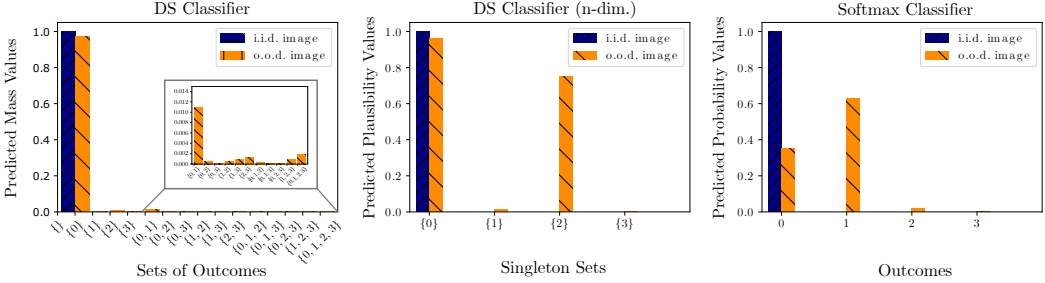

Figure 3: Predicted outputs of the DS Classifier, the n-dimensional DS Classifier and the Softmax Classifier for one illustrative example from the GTSRB dataset without (i.i.d.) and with (o.o.d.) additional noise. The true outcome for this sample is 0. Both DS approaches are more robust under increased epistemic uncertainty for this instance.

If a high level of epistemic uncertainty is prevalent during calibration via CP, a model that is incapable of capturing this type of uncertainty can be expected to frequently attribute high probability to incorrect labels. We illustrate this in Fig. 3 where we compare the outputs of the two proposed DS

Classifiers with a standard neural network classifier with softmax activation function in the output layer (denoted as *Softmax Classifier*) for one specific image from the GTSRB dataset. All classifiers are able to correctly classify the unperturbed image (i.i.d. with the training samples). In situations of high epistemic uncertainty, e.g., if the sample is o.o.d. due to additional perturbation, the Softmax Classifier assigns high probability to the wrong label while both DS Classifiers assign high mass / plausibility to the correct outcome. Especially for the $n$-dimensional DS classifier, this example emphasizes the influence of the loss function in eq. (9), which fundamentally distinguishes this approach from the Softmax Classifier (in addition to the modified activation function). Accordingly, a classifier which cannot reliably reflect epistemic uncertainty will be poorly calibratable via CP, meaning that a small $(1 - \alpha)$-quantile is obtained and prediction sets become large. We therefore argue that the obtained set size from CP can be utilized as an indication of the model's ability to learn uncertainty.

## 5 NUMERICAL EXPERIMENTS

In order to assess the capability of our method to capture predictive uncertainties, we employ it on several common datasets and benchmark it against a standard Softmax Classifier with temperature-scaling for which the CP calibration is done on the output of the true class. We conduct experiments on EMNIST (Cohen et al., 2017), GTSRB (Stallkamp et al., 2011) and CIFAR10 (Krizhevsky et al.).

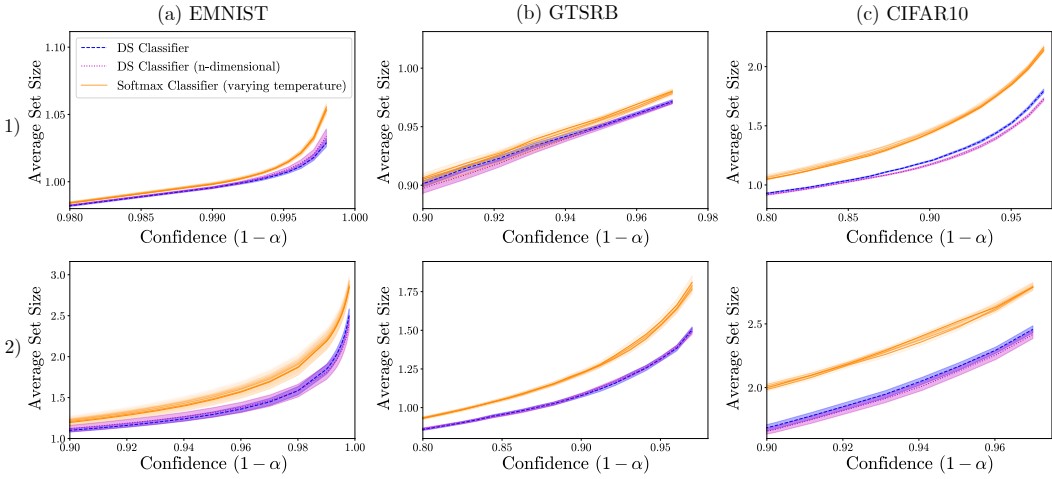

Figure 4: Average size of prediction sets resulting from conformal prediction at different confidence levels for four classes of (a) the EMNIST dataset (b) the GTSRB dataset and (c) the CIFAR10 dataset 1) without and 2) with additional perturbations applied to calibration and test images. Both of our proposed methods yield smaller or equally large sets at all confidence levels in unperturbed case, indicating more informative classifiers compared to the Softmax Classifier at different temperatures. Specifically, at large, application-relevant confidences, the gap in average set size between our approach and the reference method widens. In the scenario of enhanced epistemic uncertainty due to additional perturbation on calibration and test data average set sizes increase and the difference between the DS Classifiers and the Softmax Classifier in the set size grows, indicating that the DS Classifiers capture epistemic uncertainty more reliably.

Our initial tests were conducted on a subset of four classes from each of the above datasets to be able to efficiently apply our first method with an exponentially large output layer in the number of classes. Class selection was done such that the learning task is as difficult as possible, i.e., the selected classes are as similar as possible. For instance, for CIFAR10 we choose the subset {airplane, automobile, ship, truck}. All training details can be found in Appendix A. The hyperparameter $\lambda$ in the loss functions (8) and (9) is optimized by means of a basic line search in order to achieve the smallest average set size. We find, however, that the performance is not highly sensitive to $\lambda$, meaning that two different values, for example 0.7 and 0.8, can achieve similarly small average set sizes.

Thus, fine-tuning $\lambda$ is not required for the datasets studied. To obtain meaningful results, we analyze the average set size on the test data for all possible confidence levels $(1-\alpha)$. Note that depending on the size of the calibration dataset, not all confidence levels can be achieved, as the $(1-\alpha)$-quantile becomes inaccurate with few data and small $\alpha$. We observe that smaller (EMNIST, CIFAR10) or equal (GTSRB) average set sizes with the exponentially scaling DS Classifier as well as the n-dimensional DS classifier can be achieved compared to the Softmax Classifier with varying temperature, cf. Fig. 4. Both EMNIST and GTSRB are not considered complex datasets and therefore very small prediction sets are obtained with all methods. For the GTSRB dataset we find an average set size of smaller than one up to the plotted confidence of 97%. This can be accounted to a high test (and calibration) accuracy, which implies a $(1-\alpha)$-quantile close to 1 and hence some test samples do not have a score that exceeds it, resulting in an empty prediction set. In order to test the ability of the classifiers to capture epistemic uncertainty, we perturb images in calibration and test set

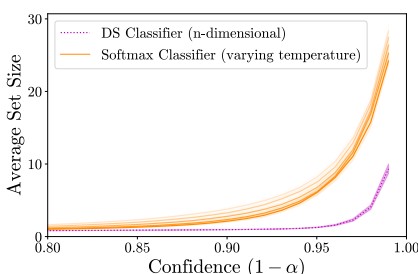

Figure 5: Average set size of the n-dimensional DS Classifier and Softmax Classifier on the full GTSRB dataset of 43 distinct outcomes at different confidence levels. The DS approach achieves significantly smaller average sets than the Softmax Classifier while retaining the same dimension of the output layer.

by randomly changing the brightness, contrast, saturation and hue of an image (GTSRB, CIFAR10) or by adding Gaussian blur with random variance (EMNIST). The models are still trained on unperturbed data, which effectively implies that calibration and test data are out-of-distribution and hence subject to higher epistemic uncertainty. In this scenario, it can be observed that the average set sizes obtained by CP are greater compared to the unperturbed case, since increased uncertainty results in the need to include more labels in the prediction set to achieve the desired confidence level, see Fig. 4 2). We find further that both DS Classifiers provide smaller, i.e., more informative average sets than the Softmax Classifier with varying temperature. Hence, one can conclude that DS approaches are able to return more reliable predictions in presence of epistemic uncertainty. To demonstrate the scalability of the n-dimensional DS Classifier, we next conduct experiments on the full GTSRB dataset with 43 classes. In situations of elevated epistemic uncertainty by perturbing images in calibration and test set, we find that the DS Classifier yields a significantly smaller average set size for all confidence levels compared to the Softmax Classifier, see Fig. 5. In particular, it is notable that the difference increases as the confidence level becomes larger. Achieving smaller average set sizes with the DS approaches compared to a Softmax Classifier with varying temperature, especially in the case of high epistemic uncertainty, supports the theoretical results of Bengs et al. (2022). Adapting the temperature, i.e., calibrating the Softmax Classifier is found not to be sufficient to reliably reflect epistemic uncertainty and obtain informative set predictions via CP. Hence, we empirically confirm the findings in Stutz et al. (2023), that CP benefits from a non-conformity measure capturing the uncertainty of the data.

## 6 CONCLUSION

Our work addresses the task of achieving trustworthy predictions in machine learning by estimating predictive uncertainty. We propose the use of classifiers based on the framework of Dempster-Shafer theory to reliably reflect not only aleatoric but also epistemic uncertainty, while still being computationally efficient during inference. By using an implicit assumption on the presence of epistemic uncertainty, we design a loss function to train the DS Classifier and formulate a reduced approach to solve the scaling issue of DST classifiers. Combined with conformal prediction as a post-processing step, we demonstrate empirically on different datasets that - compared to a standard Softmax Classifier with temperature scaling - smaller and hence more informative prediction sets are retrieved, in particular in situations of high epistemic uncertainty. Our work indicates that while conformal prediction can always satisfy coverage guarantees (by returning the trivial set), the quality of the model to capture uncertainty is crucial to obtain informative set predictions. Most of the changes that our approach entails compared to Softmax Classifiers are made in the training phase, thus DS Classifiers may prove to be widely applicable when a characterization of the underlying uncertainties is crucial.

## 7 REPRODUCIBILITY STATEMENT

Code is made available from the authors upon request. All training details can be found in Appendix A.

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

## A  DATASET AND TRAINING DETAILS

In this section, we specify the details of the experiments presented in Section 5. We utilize the Python library *PyTorch* and train all models using the Adam optimizer (`torch.optim.Adam()`). All shown results are averages of 10 runs in which the models were exposed to a different random train, calibration and test split. For all datasets we employ a CNN and spatial transformer network adapted from Hiranandani (2019). The Softmax Classifier was trained on a temperature of 1 and for calibration and test the temperature was varied between 1 and 5 in integer steps. See Section B for a discussion on the choice of softmax temperatures.

Note that the observed results to obtain smaller average set sizes by the DS Classifiers neither strongly depend on the hyperparameter $\lambda$ nor on the chosen training settings.

### A.1  EMNIST

We use the subset $\{0, 4, 8, 9\}$ of the total 10 handwritten EMNIST digits for our experiments. Training of the model is executed on $10\,000$ randomly selected samples, the calibration on $75\,000$ samples and test on $10\,000$ samples. We use a batch size of 128 and train for 35 epochs. For the first experiment with unperturbed calibration and test images the trade-off parameters yielding the smallest average set size were found to be $\lambda = 0.99$ for both DS Classifiers, indicating that the level of (epistemic) uncertainty is small.

In order to increase the epistemic uncertainty for calibration and test we apply Gaussian blur to the black-and-white images. For this, we use the built-in function from torchvision called `torchvision.transforms.GaussianBlur()` with a kernel size of $(9, 9)$ and variances in the interval $[3.0, 5.0]$. We conduct experiments on the same training settings as in the unperturbed case and adjust the trade-off parameters to $\lambda = 0.6$ for the standard DS Classifier and $\lambda = 0.7$ for the n-dimensional DS Classifier.

### A.2  GTSRB

We use the four classes

$$\left\{ \text{speed limit} \left( 20\frac{\text{km}}{\text{h}} \right), \text{ speed limit} \left( 30\frac{\text{km}}{\text{h}} \right), \text{ speed limit} \left( 60\frac{\text{km}}{\text{h}} \right), \text{ speed limit} \left( 80\frac{\text{km}}{\text{h}} \right) \right\}$$

for the initial experiments depicted in Fig. 4 1). Training of the model is executed on $1\,000$ randomly selected samples, the calibration on $3\,200$ samples and test on $1\,500$ samples. We use a batch size of 256 and train for 100 epochs. For the first experiment without additional perturbation the trade-off parameters yielding the smallest average set size were found to be $\lambda = 0.9$ for both DS Classifiers.

In order to increase the epistemic uncertainty for calibration and test we randomly change the brightness, contrast, saturation and hue of the RGB images. For this, we use the built-in function from torchvision called `torchvision.transforms.ColorJitter()` with brightness, contrast and saturation chosen uniformly from $[0.0, 3.0]$ and a hue chosen uniformly from $[-0.5, 0.5]$. We conduct experiments on the same training settings as mentioned above and adjust the trade-off parameters to $\lambda = 0.8$ for both DS Classifiers.

The experiment for the n-dimensional DS Classifier and the Softmax Classifier on the full GTSRB dataset of 43 classes was conducted on $10\,000$ train, calibration and test samples with a *ColorJitter* with with brightness, contrast and saturation chosen uniformly from $[0.5, 1.5]$ and a hue chosen uniformly from $[-0.5, 0.5]$. We use a batch size of 256 and train the n-dimensional DS Classifier for 25 epochs using the Adam optimizer and the Softmax Classifier for 40 epochs using Stochastic Gradient Descent (`torch.optim.SGD()`) with a learning rate of 0.05, a momentum of 0.9 and weight decay= of 0.0005. The hyperparameter used for the plot in 5 is $\lambda = 0.99$.

### A.3  CIFAR10

For the CIFAR10 dataset we use the subset of the four classes $\{$airplane, automobile, ship, truck$\}$. Training of the model is executed on $5\,000$ randomly selected samples, the calibration on $3\,000$ samples and test on $1\,000$ samples. We use a batch size of 256 and train for 100 epochs. For the first

experiment without additional perturbation the trade-off parameters yielding the smallest average set size were found to be $\lambda = 0.8$ for both DS Classifiers.

In order to increase the epistemic uncertainty of the CIFAR10 images for calibration and test we use *Color Jitter* as for the GTSRB dataset with brightness, contrast and saturation chosen uniformly from $[0.5, 1.5]$ and a hue chosen uniformly from $[-0.5, 0.5]$. We conduct experiments on the same training settings as in the unperturbed case and adjust the trade-off parameters to $\lambda = 0.7$ for both DS Classifiers.

## B    CALIBRATION CURVES AND EXAMPLE

Conformal Prediction allows for pre-defining a confidence level $1 - \alpha$ at which set predictions are constructed. Theorem (7) then guarantees that the coverage is at least as high as this confidence level under the mentioned assumptions. In Figure 6 we plotted the empirical coverage against the confidence for all numerical experiments in section 5 and observe that Theorem (7) is indeed fulfilled.

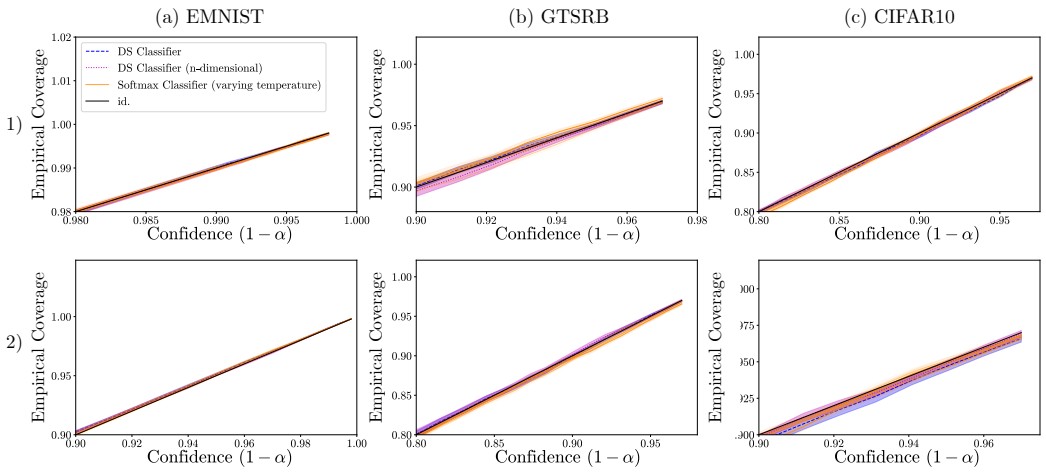

Figure 6: Empirical coverage against pre-defined confidence $(1 - \alpha)$ for numerical experiments in Fig. 4. All curves follow the identity (id.), indicating that the models are well-calibrated by Conformal Prediction.

In order to further illustrate the calibration process, it will be demonstrated in the remainder of this section in more detail. Initially, for all samples in the calibration data set, scores are determined which is the predicted plausibility of the true outcome and the predicted probability of the true outcome for the DS Classifiers and the Softmax Classifier, respectively. For the DS Classifiers, this means that the plausibility estimates of the singleton set containing the known ground truth label are used as calibration scores.

To highlight the differences between the classifiers, the scores are sorted and plotted over the data indices of the samples in the calibration set, cf. Fig. 7 for an exemplary calibration curve on the GTSRB dataset with additional perturbation on the calibration images. As shown in the plot, plausibility values of the DS Classifiers start increasing for smaller data indices than the probabilities of the Softmax Classifier for a temperature of 1 (dark orange line). Hence, at the same confidence level, larger quantiles are obtained by CP and prediction sets tend to be smaller. While predicted probabilities of the Softmax Classifier at higher temperatures increase at even smaller data indices, they tend to grow more slowly, implying that less precise predictions are obtained in general. Lower softmax temperatures would lead to a more step function shaped calibration curve, i.e., smaller quantiles for the same confidence levels and hence again larger set sizes. That is, temperature scaling of a Softmax Classifier is found not to be sufficient in presence of (high) epistemic uncertainty.

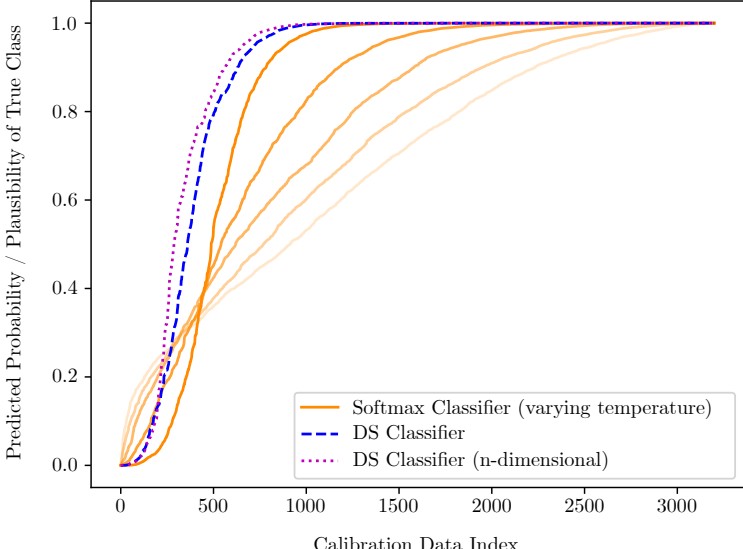

Figure 7: Exemplary calibration curve on the GTSRB dataset with additional noise on calibration data as described in A. Probability / plausibility are predicted for all samples in the calibration set, the outputs of the true label are sorted and plotted against the calibration data index. For example, the 90% quantile is the value of the respective curve at which 10% of the calibration samples have smaller predicted values. The predicted plausibilities of the DS Classifiers increase for a smaller data index than the probabilities of the Softmax Classifiers (for a temperature of 1 depicted as a dark orange line), hence a larger quantile is obtained and smaller sets can be returned. Although higher softmax temperatures start to increase for even smaller data indices, they tend to increase more slowly, indicating that less accurate predictions are obtained.

## C  METHOD OVERVIEW: COMPARISON OF THE TWO PROPOSED APPROACHES

In this section, we will illustrate the similarities and differences between the two proposed approaches based on DST introduced in section 4. See Fig. 8 for a schematic overview over the general methodologies While most of the network architecture can be chosen arbitrarily, the output layer is required to have $2^n$ dimensions for a classification problem of $n$ distinct labels in the first approach (a) and $n$ dimensions in the second approach (b). Additionally, a sigmoid activation function is used in the $n$-dimensional DS Classifier.

We train the model in (a) on the loss function presented in (8) and interpret the outputs as masses from DST. The second model in (b) is trained on the loss function in eq. (9) and we interpret the outputs as the plausibility values of the singleton sets from the DST. For this purpose, we remove the restriction of the output being normalized by using a sigmoid activation in the last layer.

In the full DS Classifier approach (a) the plausibility (and optionally belief) vector are computed in a subsequent step, which define a credal set on the probability simplex. To obtain False Negative control, we calibrate via CP on the plausibilities of the singleton sets. Accordingly, in the reduced approach (b) False Negative control is achieved without a subsequent computation step.

## D  UTILITY MAXIMIZATION

In the main text, we were able to train the DS Classifier using a custom loss function that trade-offs between a term that fits the resulting belief and one that fits the plausibility vector. This loss is motivated by interpreting this trade-off as implicit assumption on the presence of epistemic uncertainty in the data and makes use of the ability of DST to represent epistemic uncertainty in non-singleton sets. Here, we sketch a different route to train DS Classifiers which is tied to utility maximization instead and demonstrate how similar behavior is obtained also for this approach.

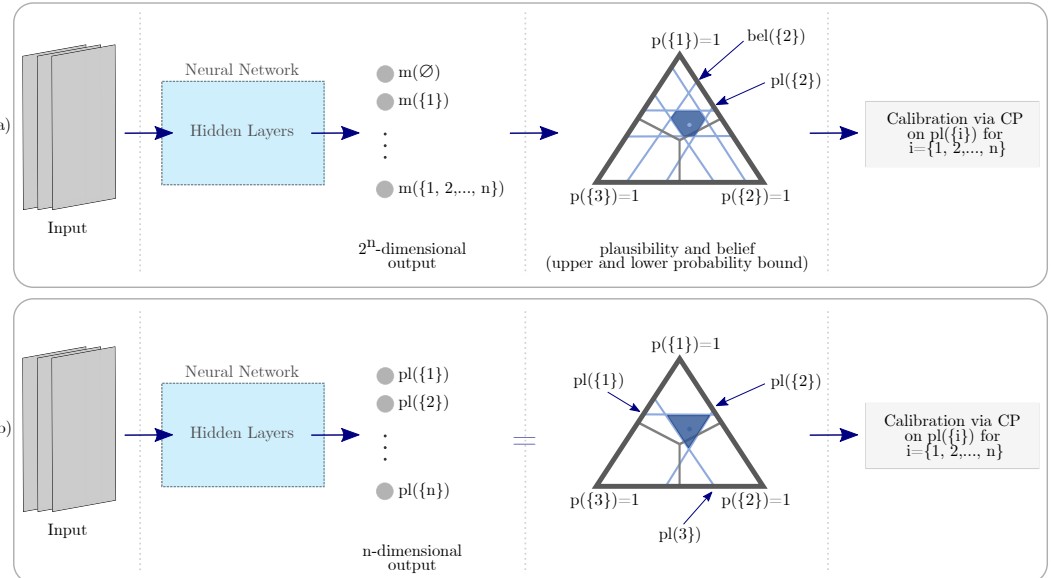

Figure 8: Overview of the two proposed methodologies. We employ a neural network with arbitrary architecture, only requiring an adapted output dimension (a) or a sigmoid activation function in the output layer (b). We interpret the output of the network in the first approach to be the masses from DST, and in the second approach to be the plausibilities of the singletons. After training on a suitable loss function, e.g., (8) in (a) and (9) in (b), a calibration according to CP is performed on the plausibilities of the singleton sets of the true labels to obtain set-valued predictions with confidence guarantee.

In utility maximization (Zaffalon, 2002; Corani & Zaffalon, 2008; 2009; Yang et al., 2017), a score is given to different prediction outcomes, which may overlap. For the classification setting investigated here, the frame of discernment of DST, the powerset of all possible class combinations, contains the prediction outcomes to be considered by utility maximization. Having the same form, we could re-interpret the prediction vector of the DS Classifier as utility score vector rather than mass vector. Correspondingly, the target vector we want to fit to and the loss function we employ need to be adopted as well:

Where the DS Classifier is pushed towards more or less uncertain predictions by the loss, utility maximization assigns smaller or larger utility to the corresponding sets. Hence, to recast the implicit assumption in presence of epistemic uncertainty previously done in the loss construction, we now simply put smaller (larger) target utility values on larger prediction sets for a smaller (higher) presence of epistemic uncertainty. The assumption now being hidden in the target vectors, the loss function can be any classical distance loss, e.g., a mean squared error or cross-entropy loss. Target utility scores can be user-defined in a highly application specific setting see, e.g., Hüllermeier & Waegeman (2021). For classifiers we obtain good results using the quadratic utility score function $u_{\delta\gamma}$ (Zaffalon et al., 2012) with standard parameter settings $((\delta, \gamma) = (1.6, 0.6))$.

The classifier is then trained to maximize the utility over the training data, which has an uncertainty-based interpretation. In a stretch of theory, we can now take this output vector and again interpret it as if it would be a DST mass vector and process it further equivalently to the procedure in the main text.

To demonstrate the similar behavior of the thus constructed *utility DS Classifier*, we show the average set sizes as a function of confidence levels from CP with or without additional perturbations on EMNIST, GTSRB and CIFAR10 datasets (equivalently to Fig. 4), see Fig. 9. The setup of the experiment follows the main text, we report the results here only.

As can be seen in Fig. 9, the utility-label trained DS Classifier closely follows the DS Classifier trained using the trade-off loss and obtains similar improvements compared against standard softmax

classifiers. We conclude that both construction approaches are useful and offer two routes to interpret the benefits of DS Classifiers.

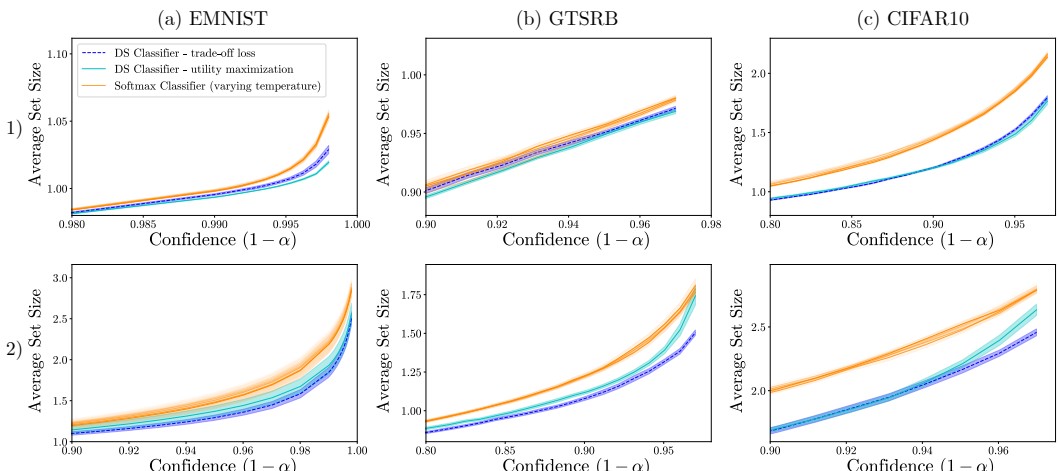

Figure 9: Average set size on different datasets 1) without and 2) with additional perturbation in calibration and test dataset for the DS Classifier and Softmax Classifier as presented in the main text and the *utility DS Classifier* introduced here. The utility DS Classifier closely follows the DS Classifier trained using the trade-off loss and obtains similar improvements compared against the Softmax Classifier.

# E  SUB-NORMALIZATION OF MASS FUNCTIONS

Dempster-Shafer theory requires mass functions to be normalized (cf. eq. (2)) and the mass of the empty set to be zero (cf. eq. (1)). This section is aimed at examining to what extent both conditions are fulfilled in our proposed methods.

In the full DS Classifier, normalization of the mass function is guaranteed by a softmax activation function in the output layer and the design of the loss function (eq. (8)) enforces an (approximately) zero mass on the empty set after training. Fig. 10 confirms this for test data from numerical experiments on the GTSRB data set from section 5. Both training on i.i.d. and o.o.d. data yield predicted masses of the empty set which do not exceed $10^{-2}$ and hence mass functions satisfy both conditions to a very good degree of approximation.

For the $n$-dimensional DS Classifier normalization is not explicitly enforced during training. As this approach does not provide the mass function itself but only the plausibilities of the singleton sets, we first note that sub-normalization indirectly corresponds to plausibilities of singleton sets which sum up to less than one (in this case eq. (4) can only be fulfilled if mass is assigned to the empty set). As Fig. 10 depicts, only a very small fraction of the test set exhibits a summed plausibility of less than one. That is, the reduced DS Classifier also yields (approximately) normalized mass functions. Given the observed similar behavior of the reduced approach to the full DS classifier, this suggests a small impact of few subnormalized predictions.

We further note that easy remedies exist to gain guaranteed normalized outputs: First and trivially, the zero set can be pruned from the output layer, thus guaranteeing a zero mass value for the empty set and thus normalized mass predictions for all other sets. Second, one can slightly re-interpret the empty set as so called conflict $K$, occurring for example when combining two mass vectors according to DST, what allows an interpretation of o.o.d. behavior for the current prediction. DST, more specifically the rule of combination (Shafer, 1990), then simply applies a scaling factor of $\frac{1}{1-K}$ to all masses to re-obtain normalized predictions. Note that this rule is known to produce instable behavior in cases of very high conflict (since it amplifies small predicted masses by a large factor), which we do not expect to encounter here given our empirical results.

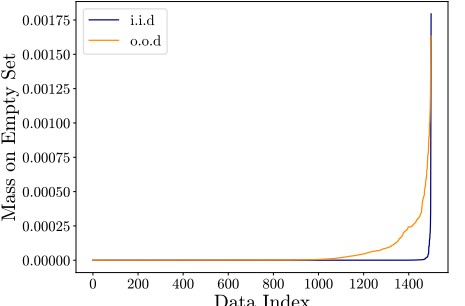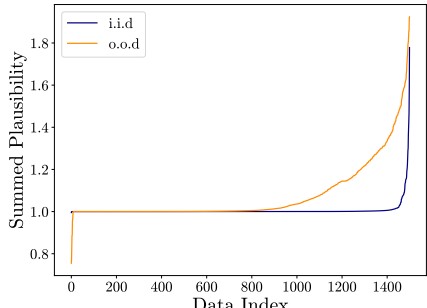

Figure 10: Exemplary plots for the GTSRB data set without (i.i.d.) and with additional perturbation (o.o.d.) on the training data. It can be observed that the masses of the empty set resulting from the DS Classifier are strictly smaller than $10^{-2}$ for all samples in the test set, indicating that mass functions approximately fulfill eq. (1). For the $n$-dimensional DS Classifier a summed plausibility smaller than 1 corresponds to a non-zero mass of the empty set. That is the case for only a small fraction of the samples and hence also the reduced approach approximately satisfies required conditions.

For the reduced DS classifier neither of these remedies is accessible directly since the empty set is not present in the output layer. However, an additional working assumption that for cases in which subnormalization occurs $pl(A_i) = m(A_i)$ ($A_i$ being a singleton set prediction). This allows us deduce the empty set mass as $K = 1 - \sum_i m(A_i)$ and then proceed with the rescaling with $\frac{1}{1-K}$, as before.

## F  TOY EXAMPLE ILLUSTRATING EFFECT OF EPISTEMIC UNCERTAINTY ON DS CLASSIFIERS

In addition to the numerical experiments on image data conducted in section 5, this section aims to highlight the impact of epistemic uncertainty on DS and Softmax Classifiers by using an illustrative data set. We consider instances $x$ that are sampled from Gaussian distributions and say that $x$ belongs to class $i$ if $x \sim \mathcal{N}(\mu_i, \sigma_i)$. The goal is to correctly predict classes $i$ given inputs $x$.

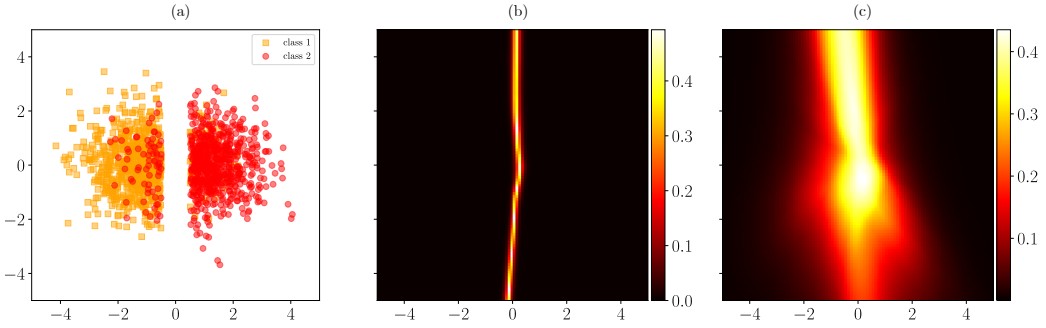

Figure 11: Training instances from class 1 that are sampled from $\mathcal{N}(-1, \mathbb{I}_2)$ (light orange squares) and instances from class 2 sampled from $\mathcal{N}(+1, \mathbb{I}_2)$ (dark orange circles). In order to increase epistemic uncertainty, a margin in one dimension is added within which no training data is available (a). Minimal softmax value obtained from a standard Softmax Classifier on test data (b) and mass of the full set $\{1, 2\}$ obtained from the DS Classifier on test data, reflecting epistemic uncertainty of both classes (c). While the Softmax Classifier learns a sharp decision boundary in regions where no training data is available, DS Classifiers capture epistemic uncertainty more reliably by leaning a high mass on large sets.

Our first experiment is a classification problem of two distinct classes in two dimensions (cf. Fig. 11 (a)). We train a Softmax Classifier as well as a DS Classifier on a training set that has samples of regions of high overlap removed (no samples within a certain margin). This aims at increasing the level of epistemic uncertainty and provoking a setting in which the Softmax Classifier is likely to yield overconfident behavior. Our experiments strengthen this assumption as is reflected in Fig. 11 (b), where the minimal softmax value is shown. Only a narrow region in between the two distributions has high minimal softmax values (around 0.4), indicating that the classifier is overconfident elsewhere. Predicted masses of the total class set $\{1, 2\}$ of the DS Classifier is high in a broader region around the overlap of the distributions, indicating that it learned epistemic uncertainty more reliably and yielding less overconfident predictions (note that in a two-label setting epistemic uncertainty of both classes is given by $pl(\{1\}) - bel(\{1\}) = pl(\{2\}) - bel(\{2\}) = m(\{1, 2\})$.

Following procedures from section 5, calibration of this problem is done on the full sample set (without removing samples within the margin). We show obtained predicted probabilities and plausibilities as well as belief values in Fig. 12. Here, overconfidence of the Softmax Classifier is reflected by a high fraction of samples in the calibration set with a predicted probability of almost zero. Plausibilities obtained from the DS Classifier increase more gradually and hence yield more conservative predictions.

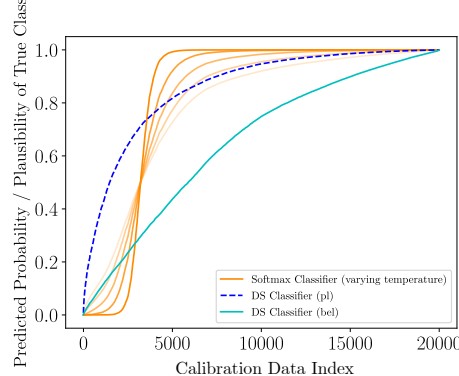

Figure 12: Calibration curves for the Softmax Classifier at different temperatures (orange) and the DS Classifier. In addition to the plausibility, which is used for calibration of the DS Classifier, the resulting belief values obtained from the predicted mass are displayed. While Softmax Classifiers misclassify a large fraction of calibration data (low predicted probability) in presence of epistemic uncertainty, predicted plausibilities obtained from DS Classifiers increase more gradually.

Applying Conformal Prediction as a post-processing step on both classifiers at different confidence levels $(1 - \alpha)$ provides prediction sets. We display average set sizes on test data of both approaches in Fig. 13 (a). For a classification problem on Gaussian data with only two classes no difference in set size can be observed. This is, however, in agreement with findings from section 5, where less complex problems like EMNIST show smaller differences in average set size, while more complex data sets as CIFAR10 exhibit larger performance gaps between DS and standard approaches. In order to confirm these findings on Gaussian data, larger classification problems with 10 and 20 different classes are performed for the $n$-dimensional DS and Softmax Classifier, cf. Fig. 13 (b) and (c), respectively. Indeed, in such higher dimensional settings differences in average set sizes increase. Gaussian distributions from which instances are sampled have 5 (b) and 10 (b) dimensions and both models are trained on data where samples within a margin are removed in order to increase epistemic uncertainty. We construct the Gaussians in such a way that in each two new dimensions two Gaussians are placed at -1 and +1 in the first dimension and the margin is along the second new dimension. Variances in all dimensions are set to one.

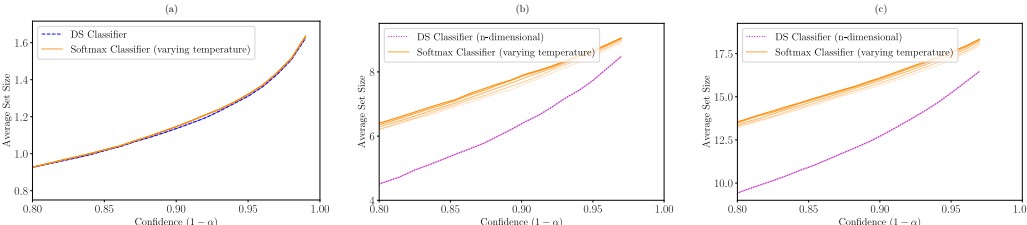

Figure 13: Average set size obtained via CP at different confidence levels $(1 - \alpha)$ for 2 (a), 10 (b) and 20 (c) classes of Gaussian data, i.e., instances sampled from 2, 10 and 20 different Gaussian distributions of dimension 2, 5 and 10, respectively. Samples in regions of high overlap of the Gaussians is removed during training in analogy to Fig. 11 (a). While for the smallest problem of two Gaussian distributions no differences in set size are observed between the DS Classifier and the Softmax Classifier, increasing performance differences between the $n$-dimensional DS Classifier and the Softmax Classifier become evident. This result is in agreement with the result shown in Fig. 4, where only small differences in set size are obtained for less complex problems like EMNIST, but larger gaps occur for complexer data sets (e.g., CIFAR10).

