# OpenReview forum: "Reliable Classifications with Guaranteed Confidence using the Dempster-Shafer Theory of Evidence"
_ICLR.cc/2024/Conference — Submitted to ICLR 2024_

### Official Review · Reviewer_JFXw · 2023-10-29

**Soundness:** 2 fair
**Presentation:** 2 fair
**Contribution:** 1 poor
**Rating:** 3
**Confidence:** 5

**Summary:**

This paper proposes to perform the learning of calibrated belief functions, through the combination of an evidential classifier and conformal prediction. Adapted loss functions are proposed, and experiments show that the set size for a given confidence level is better for DS approaches.

**Strengths:**

+: an potentially important topic, which is combining the assessments of various facets of uncertainty with the idea of calibration, or in other words how to obtain well founded estimation of both so-called epistemic and aleatoric uncertainties.

+: rather well-written as a whole.

**Weaknesses:**

While the topic covered by the paper is important, I found that the paper was smeared with approximation, and not very clear about its technical content (to the point that I would not be able to reproduce the experiments, should I want to). This maybe due to the space restrictions, but not only (and even that would just mean that the paper should be done for a venue allowing for longer version, i.e., Journals or ArXiv): indeed, even with the appendices it would be hard to answer some questions/reproduce the methods/results, and some approximations made in the paper are not due to space limitations. Below are some more detailed comments. When those could lead to a potential comment/rebuttal from the authors, I have put a star (*) next to them (so that they can be considered questions to answer).

Detailed comments:

- (*)Semantic positioning: the authors are not always really clear as their interpretation of belief functions, and this should definitely be clarified, as in the current work the semantic and the resulting belief functions can have an important impact in practical application. for instance, it is suggested in the manuscript that obtained belief functions may be subnormalised, but it is also mentioned that decision rules issued from a probability set interpretation could be used to deal with decision oriented problems. Clearly, these two statements (without further clarification about the positioning) are logically inconsistent, as one cnanot use the imprecise probabilistic interpretation with sub-normalised BF.

- (*)Focus on recent works on BF published in ML venues: the authors mention (multiple times) that BF was only recently applied to ML. This is rather true if authors means "have recently appeared in ML focused top-tier venues", but rather untrue if authors consider ML and statistical learning/inference as a field (and do not filter by venues). There is a huge literature on learning belief functions, and I would even argue that this is one of the main topic (with information fusion) on which belief functions has been applied (as opposed to other uncertainty theories, e.g., possibility theory that has mainly been considered for logical reasoning). The same is true for the "uncertainty quantification" community mentioned in P3, especially since "uncertainty quantification" or UQ for short covers a large group including classical risk analysis. My perception is that authors mean "Uncertainty quantification in ML top-tier venues for the past 5 years"... which is rather restricted as a span, IMHO.

- (*)Lack of connection with potentially relevant streams of work: I would say that the current proposal should at least make a clear positioning with respect to two lines of work: the first one is about obtaining calibrated belief functions, a topic currently championed by Ryan Martin (who recently linked his work to CP), and the second one is that of adapting loss functions to credal labels, i.e., labels described by a probability set (see recent works, including some published in top-tier AI venues, by Julian Lienen on this topic).

- (*)P1: authors mention instance-risk wise control, yet it is known (see "The limits of distribution-free conditional predictive inference") that obtaining full conditional coverage in a distribution-free setting is impossible. Whether authors are chasing that should be specified.

- (*)P1: the part about epistemic/aleatoric is a bit loose. Imprecise data can definitely belong to epistemic uncertainty (and could be reducible in principle, as well as noisy data in the case where better sensors/measurement tools can be found), and I would question the idea that "non-optimal training" or "ill-chosen" hypothesis can be reduced by obtaining more data (even an infinite amount of data would not allow to change the hypothesis space, nor the fact that a learning procedure is sub-optimal).

- P2: I am a bit skeptical about the use of set-vaued predictions in real-time setting, as set-valued predictions typically beg for a post-processing of some kind, rather than e.g., pessimistic decisions that can be directly plugged in uncertainty estimates. So the argument/connection looks at least a bit weak/irrelevant to me.

- P3: it is strange to cite reject (Herbei/Wegkamp) just after a plea for conformal approaches, as if I am correct, the reject option proposed in this paper does not deal with calibration?

- P4: strictly speaking, a Bayesian approach would put a prior over every possible proability values (typically a Dirichlet distribution), who would be uniform in case of no experiments, and rather skewed in case of presence of observations. The critic done there rather corresponds to the need to consider second-order models (mention after by the authors), of which Bayesian approaches constitute an instanciation. So here again, I would say that the argument is not very well crafted.

- P5: while I agree that the full class set is always conservatively valid, it is not strictly valid in the sense sought by conformal prediction, that aims at turning equation (7) into an equality.

- (*)Whole section 4: this whole section is not detailed enough so as to make the whole approach reproducible (and a look at the appendices indicate that the information cxannot be found in them either). For instance, I cannot really understand from the text 1. what is the quantity that is conformalised and 2. what are the used scores to conformalise it. In particular, if the conformalised quantities are the belief function (as I think it is), how are obtained the necessary ground-truth allowing to guarantee calibration? What is the random quantity against which we conformalise? What are the links between these loss functions and the recently introduced credal loss functions?

- (*): P7: the claim that using belief could lead to a control of false positive would need to be exposed much more lenghtly. The connection is definitely not direct for the average reader, and I would say also for an expert reader.

- (*)Experiments: experiments mainly shows that the proposed approach results in sets of smaller sizes, however there are at least two critics about them. The first is that they do not compare to all the recent works aiming at producing set-wise predictions (and referenced by the authors), the second is that it is not possible to find in the current publication (including in the appendices) graphs showing whether the proposed methods do actually produce caibrated predictions (in the sense of Equation (7)).

**Questions:**

See weaknesses.

**Details Of Ethics Concerns:**

No need here

---

> ### Author Response · Authors · 2023-11-20
> **Response 1/3**
>
> We thank the reviewer for taking the time to carefully read our manuscript and for valuable suggestions that help us improve our paper. We appreciate the literature suggestions that help us better put our work into context. In the following we respond to the raised questions / remarks one by one.
>
> >the authors are not always really clear as their interpretation of belief functions, and this should definitely be clarified, as in the current work the semantic and the resulting belief functions can have an important impact in practical application. for instance, it is suggested in the manuscript that obtained belief functions may be subnormalised, but it is also mentioned that decision rules issued from a probability set interpretation could be used to deal with decision oriented problems. Clearly, these two statements (without further clarification about the positioning) are logically inconsistent, as one cnanot use the imprecise probabilistic interpretation with sub-normalised BF.
>
> We construct belief and plausibility values from the predicted mass values according to the (standard) construction rules in eqs. (3) and (4). We included an additional cross-reference  in the method section to remove any remaining unclarity.
> As to the topic of subnormalisation, we were not explicit about it in the paper for two reasons: It rarely occurs and it is easy to deal with. We included an additional appendix to address subnormalisation and make the presentation of the method more complete. Here a synopsis of our answer:
> For the $2^n$ dimensional DS classifier, training enforces the classifier to output (approximately) 0 mass for the empty set. Across our experiments, we didn’t encounter data set with any substantial ($>10^{-2}$) amplitude in the empty set. To safeguard the approach against the possibility of empty set predictions, there are several mitigation strategies: First, one can simply reduce the output layer by the zero-set dimension. Second, one can slightly extend DST and interpret the empty set as OOD behavior, interpret it as conflict K and normalize the remaining mass outputs according to DS rule of combination (i.e., scale by $\frac{1}{1-K}$).
> For the reduced classifier, subnormalisation occurs more easily, though still rarely (see Figure 10). Since the output is here interpreted as singleton set plausibilities, subnormalisation indirectly implies mass assignment to the empty set. With an additional assumption that for cases where subnormalisation occurs the singleton set plausibility equals the corresponding mass estimates, a rescaling as outlined above can easily be done. In our experiments, however, we don’t expect a significant impact due to the low occurrence frequency of subnormalisation.
>
> >Focus on recent works on BF published in ML venues: the authors mention (multiple times) that BF was only recently applied to ML. This is rather true if authors means "have recently appeared in ML focused top-tier venues", but rather untrue if authors consider ML and statistical learning/inference as a field (and do not filter by venues). […] I would say that the current proposal should at least make a clear positioning with respect to two lines of work: the first one is about obtaining calibrated belief functions, a topic currently championed by Ryan Martin (who recently linked his work to CP), and the second one is that of adapting loss functions to credal labels, i.e., labels described by a probability set (see recent works, including some published in top-tier AI venues, by Julian Lienen on this topic)
>
> It is indeed regrettable that DST and works on belief functions (as many other methods!) only slowly diffuse from the statistics community to the machine learning community. In fact, the comments of the reviewer him-/herself and reviewers 1 and 3 in extension highlight the need for more ML publications exploring these (in the ML community at least) understudied concepts, as we do here. We are optimistic that the repertoire of today’s ML capabilities will be extended by new approaches resulting from study of the intersection of these domains.
>
> The aim of our literature overview is not to extensively review work on belief functions, but to provide necessary background to show the context in which this work was developed and provide interested readers with the links necessary to read up on background information. We extended the literature overview to also encompass more recent works on DST / BFs in ML.

---

> ### Author Response · Authors · 2023-11-20
> **Response 2/3**
>
> >authors mention instance-risk wise control, yet it is known (see "The limits of distribution-free conditional predictive inference") that obtaining full conditional coverage in a distribution-free setting is impossible. Whether authors are chasing that should be specified.
>
> The main contribution of our work is a training procedure for DS classifiers and a reduced approach for scaling to larger problems, which enables a lightweight classifier with intrinsic capability to represent epistemic and aleatoric uncertainty simultaneously. We then employ conformal prediction to construct set predictors with known confidence demonstrating improvements by using the proposed approach. As such, no claim towards full conditional coverage in a distribution free setting is made.
>
> >the part about epistemic/aleatoric is a bit loose. Imprecise data can definitely belong to epistemic uncertainty (and could be reducible in principle, as well as noisy data in the case where better sensors/measurement tools can be found), and I would question the idea that "non-optimal training" or "ill-chosen" hypothesis can be reduced by obtaining more data (even an infinite amount of data would not allow to change the hypothesis space, nor the fact that a learning procedure is sub-optimal).
>
> We follow the definitions of Hüllermeyer & Waegeman (2021) for the definitions of aleatoric and epistemic uncertainty. In the revised version we sharpened the corresponding formulation to avoid confusion.
>
> >I am a bit skeptical about the use of set-valued predictions in real-time setting, as set-valued predictions typically beg for a post-processing of some kind, rather than e.g., pessimistic decisions that can be directly plugged in uncertainty estimates. So the argument/connection looks at least a bit weak/irrelevant to me.
>
> Such pessimistic decisions occur when the risk for one of the set members is extremely high. Obviously, a set prediction is not a decision or a system acting on it. There is some application dependence here, which cannot be resolved in general.
>
> >P3: it is strange to cite reject (Herbei/Wegkamp) just after a plea for conformal approaches, as if I am correct, the reject option proposed in this paper does not deal with calibration?
>
> We cite Herbei / Wegkamp as part of the literature review showing related methods, there is no criticism or rejection of their approach in the manuscript. We are thus a bit confused what the reviewer means by “cite reject”? It is separated from the discussion of works on conformal prediction semantically (from the paper: [foregoing discussion of set predictors] “The first one is [citations of conformal prediction works]. […] the second branch, […]. A special case of this is [Herbei & Wegkamp citation]”). We hope this clears up any remaining confusion.
>
> >P4: strictly speaking, a Bayesian approach would put a prior over every possible proability values (typically a Dirichlet distribution), who would be uniform in case of no experiments, and rather skewed in case of presence of observations. The critic done there rather corresponds to the need to consider second-order models (mention after by the authors), of which Bayesian approaches constitute an instanciation. So here again, I would say that the argument is not very well crafted.
>
> Second order models in general can capture epistemic (as well as aleatoric) uncertainty. However, necessary computational resources / latency times must be considered as well when crafting applications using such methods. The overlap of these two requirements in fact motivates our work – we point the reviewer to the corresponding paragraph in our introduction, which also includes Bayesian networks in its discussion.
>
> >P5: while I agree that the full class set is always conservatively valid, it is not strictly valid in the sense sought by conformal prediction, that aims at turning equation (7) into an equality.
>
> We included a specification of our usage of the term “valid” in order to avoid confusion.

---

> ### Author Response · Authors · 2023-11-20
> **Response 3/3**
>
> >this whole section is not detailed enough so as to make the whole approach reproducible (and a look at the appendices indicate that the information cxannot be found in them either). For instance, I cannot really understand from the text 1. what is the quantity that is conformalised and 2. what are the used scores to conformalise it. In particular, if the conformalised quantities are the belief function (as I think it is), how are obtained the necessary ground-truth allowing to guarantee calibration? What is the random quantity against which we conformalise? What are the links between these loss functions and the recently introduced credal loss functions?
>
> We invite the reviewer to re-evaluate sections 4/5 (see second paragraph after equation 9) and appendices A and especially B.
>
> What is the quantity that is conformalised? – In section 4 we discuss how plausibility values are being used as calibration scores (and not belief values!). To be even more explicit, we dedicated appendix B to the calibration procedure. Calibration curves for all three approaches (full and reduced DS classifier, softmax classifier) are shown and explicitly identified as plausibility scores, as is written in the accompanying text.
>
> Concerning the question about ground truth values needed for calibration, these are easily accessible in our setting, as we deal with single-class classifiers. The classification settings are described in detail in the main text (section 5) and Appendix A. Generation of calibration scores can be found in Appendix B. Here we added an additional phrase bridging the nomenclature to set prediction papers, so that no confusion can arise.
>
> >P7: the claim that using belief could lead to a control of false positive would need to be exposed much more lenghtly. The connection is definitely not direct for the average reader, and I would say also for an expert reader.
>
> In a format with restricted space as in the ICLR, it is not possible to discuss other utilizations of proposed methods at length. The sidenote of the connection of belief values as lower bound on the probability to false positive control stands in no connection to the main body of the text and, as the reviewer pointed out, may confuse readers. As such, we substitute the original formulation with a more general statement expressing possible other utilizations of credal set predictions, which extend beyond the demonstrations of the paper .
>
> >Experiments: experiments mainly shows that the proposed approach results in sets of smaller sizes, however there are at least two critics about them. The first is that they do not compare to all the recent works aiming at producing set-wise predictions (and referenced by the authors), the second is that it is not possible to find in the current publication (including in the appendices) graphs showing whether the proposed methods do actually produce caibrated predictions (in the sense of Equation (7)).
>
> Considering the first point: As stated above and in the answers to the other reviewers, the main contribution of this paper is really the construction / training of DS classifiers and a scaled (reduced) version for lightweight uncertainty estimation, which is useful for precise set estimation under uncertainty. Referring to the answers of reviewers 1 and 3, we included an additional appendix showcasing the developed approach on toy data to further highlight its behavior also without calibration procedure. Further experiments on different applications utilizing epistemic uncertainty estimates (e.g., active learning) as well as in-depth discussions of connections to / benchmarking against existing set predictors are left to future work.
>
> Considering the second point: With a rich literature of conformalized classifiers already in place, we felt that it would be a bit repetitive to put another plot showing that CP indeed achieves calibration coverage according to eq. (7). Nevertheless, we now included a plot showing calibratedness on our examples in Appendix B.

---

> > ### Comment · Reviewer_JFXw · 2023-11-23
> > **Thanks for the feedback + sorry for the lack of time to interact**
> >
> > Dear authors,
> >
> > Thanks for the detailed feedback and answers, as well as for the minor modifications in the main body of the paper. Unfortunately I am currently involved in lab evaluation committees which take up to ten hours of work daily, and does not have the time to really engage in time-constrained, online discussions (such things are better suited to journal publications that are less time-pressured). I apologize for this, but will take a closer look at the answers and changes to make a final recommendation.
> >
> > (quickly) looking at your new graphs in the appendix, I am really wondering about the plus-value fo the approach compared to just take the normalized p-values of classical conformal prediction interpreted as possibility distributions (see works of Martin and Cello on this)? I am also wondering to which extent the considered data sets are not "easy" calibration-wise, as a simple temperature scaling on a softmax seems to give quite satisfactory results in terms of claibration?
> >
> > Best regards

---

> > > ### Author Response · Authors · 2023-11-23
> > >
> > > We completely understand the time pressure caused by the review process besides all other research activities… Here a quick answer to the questions:
> > >
> > > @”are data sets easy to calibrate?”: In general no, although the 2D setting of the toy data probably is. To answer this question, we included temperature scaled calibration curves and set-size vs. confidence curves in all corresponding plots. Our findings indicate that a temperature scaling only marginally influences set sizes of an uncalibrated softmax classifier on any data. Smaller set sizes by DS Classifiers already occur in settings as easy as high dimensional gaussians (see Fig. 13). On calibration curves we see that while one can tune the calibration scores to be more conservative on the left (i.e., uncertain) side of the calibration curve via temperature-scaling, this goes on cost of lower scores for correctly classified data points, which, in turn, seems to deteriorate set prediction performance. It appears a temperature scaling cannot turn an aleatoric uncertainty estimate into an estimate for epistemic uncertainty.
> > >
> > > @”can we just take p-values of CP and interpret them as possibility values?”: Yes, but we would like to have a good estimate of epistemic uncertainty in order to have tight scores (not unnecessarily loose possibility estimates). Consider a constructed pathological example, the maximally (epistemically) uncertain classifier, i.e., the classifier that simply returns a random probability estimate for all classes for any data point. We can still calibrate this classifier by using CP (although we would obtain rather large set sizes), the p-values and possibility values would be rather pessimistic estimates. Now consider a 2nd order classifier that estimates its uncertainty perfectly. In case of no epistemic uncertainty, we would obtain the correct probability estimate and could simply read out the plausibilities / possibilities. In case of epistemic uncertainty, this ideal classifier would “just” include the true probability estimate and we would obtain a tight, minimally small possibility / plausibility. When constructing set predictors by CP,  we would hence never include any unnecessary label in our prediction set for a given confidence. The work from Martin nicely shows this link between CP and possibilities, but possibilities estimated by interpreting p-values such can be quite poor (i.e., loose) if epistemic uncertainty is not estimated well. Our work proposes DS Classifiers as cheap epistemic uncertainty estimators which improves CP (in terms of set sizes).

---

### Official Review · Reviewer_FfYh · 2023-10-30

**Soundness:** 3 good
**Presentation:** 2 fair
**Contribution:** 2 fair
**Rating:** 5
**Confidence:** 4

**Summary:**

The paper introduces a new reliable set-valued classification approach. It is based on the Dempster-Shafer Theory, which aims to train a prediction function mapping from the sample space to a set of class labels (with dimensional $2^n$ where $n$ is the number of classes). A conformal prediction procedure is then applied to the so-called "plausibility" of singleton set of the ground true label to obtain the set-valued predictions.

**Strengths:**

The framework that the paper studies is new and has been understudied.

**Weaknesses:**

There is no study of the theoretical properties of the proposed method.

Some claims are not fully justified (see questions below.) This may be improved with a better presentation and an ablation study.

Some presentations are unclear (see questions below.)

**Questions:**

1. On page 6, section 4: It stated "Our method promotes a basic classifier of any kind into a probabilistic set predictor h: X →2^Y that outputs a mass vector from DST, cf. Fig. 2. The function ˆh is expected to have the property that it assigns higher mass to larger sets for instances x with high epistemic uncertainty. In cases of high aleatoric uncertainty, ... For low predictive uncertainty, ..." How to enforce an off-the-shelf machine learning classifier to have these properties? Moreover, how to make sure that the resulting function has the property that $h(A)\le h(B)$ when $A\subset B$? It seems that a basic classifier has to be tailored to achieve this property. Later on it stated that the approach "is applicable to arbitrary network architectures". I am afraid that I am entirely sold on this.

2. On page 7, it stated that "Here, we propose a restriction useful for the common case of false negative control by interpreting
the outputs of the model as the plausibilities of the singleton sets directly and hence do not need to compute them in a post-processing step." I am confused on two fronts.

    2.1 The loss function in (8) involves the plausibility and belief outputs, which are computed from the mass (output of the basic classifier). To update the network, the gradient of the loss function has to be computed, which necessarily have to take the mass to the plausibility/belief computation into consideration. So it is not really a post-processing step, but rather a fairly integral step. Correct me if I am wrong.

    2.2. For the second approach, it stated that "The only adjustment to standard models in our second approach is that outputs are not normalized to 1." But later on the loss function is replaced as well. Do you mean that the basic classifier still has a $2^n$ dimensional output, but only $n$ components are used in the loss function (the rest are discarded), or do you mean that basic classifier has an $n$ dimensional output to begin with? If it is the latter case, then the difference between the two approaches are more substantial. Moreover, the second approach would not be related to the Dempster-Shafer Theory at all.

Moreover, if the "post-processing step" is removed for the second approach, then an updated graphic representation is needed in addition to  Figure 2, instead of the two approaches sharing the same figure. A dedicated figure may help clarify any confusion.

3. More to the second approach: I do not quite understand the role of the $\lambda$ parameter in loss function (9). Shouldn't both CE and MSE have a somewhat same/similar goal? If $\lambda=0$, then wouldn't the second approach reduce to a typical classification method? In this case, the only novelty in the second approach would be a half-new loss function with the CP in the end.

4. The procedure ends with the CP applied to the plausibility. Here the plausibility is used merely as a conformity score. One can't help wonder if the result is due to CP or due to the choice of the score. Can we achieve similar performance if CP is applied to the softmax score or any other score of a standard $n$-dimensional classifier ($n$ is the number of classes)? In reverse, an ablate study is needed to see how the methods perform without the CP method in the end.

---

> ### Author Response · Authors · 2023-11-20
> **Response 1/3**
>
> We thank the reviewer for carefully reading our paper and for detailed questions and helpful suggestions to further improve the presentation of our method. In order to address the questions concerning differentiation of full and reduced approach, we have updated Fig. 8 (former Figure 7) and Appendix C as a whole for better clarity and direct comparison between the full and reduced approach. Here are our responses to the other concerns and questions:
>
> Regarding 1.:
> > On page 6, section 4: It stated "Our method promotes a basic classifier of any kind into a probabilistic set predictor h: X →2^Y that outputs a mass vector from DST, cf. Fig. 2. The function ˆh is expected to have the property that it assigns higher mass to larger sets for instances x with high epistemic uncertainty. In cases of high aleatoric uncertainty, ... For low predictive uncertainty, … How to enforce an off-the-shelf machine learning classifier to have these properties?
>
> Our approach enforces the desired properties by the suggested training procedure. I.e., it demands that the output layer of a basic probabilistic classifier (e.g., a standard NN with softmax activation in the output) is adapted. While off-the-shelf ML classifiers typically have $n$ output dimensions for an $n$-dimensional classification problem, the DS classifier proposed in our work requires to replace such an output layer by a $2^n$-dimensional output. That is done to match the number of elements in the powerset $2^\Theta$ where $\Theta =  \{1, 2, …, n\}$ are the $n$ distinct labels or classes. In order to obtain meaningful probabilities or “masses” as they are denoted in DST to all $2^n$ outputs and therefore enforce the classifier to have the mentioned properties (high epistemic unc. -> high mass on larger sets, high aleatoric unc. -> equally distributed mass across singleton sets) we introduce the loss function in eq. (8) with adjustable parameter $\lambda$ that trades-off putting high mass on larger sets (small $\lambda$) and putting high mass on smaller sets (large $\lambda$). In that way, $\lambda$ can be seen as an optimizable hyperparameter that reflects the degree of epistemic uncertainty.
>
> > Moreover, how to make sure that the resulting function has the property that h(a) \leq h(B) when A \subseteq B?
>
> In our first approach we interpret the resulting function h(A) as mass function from DST, that only requires the mass function to be normalized, which we guarantee by using softmax activation on the output layer and the mass of the empty set to be zero, which is enforced by labels and the corresponding loss function. Since we want the DS Classifier to be able to express epistemic uncertainty by putting high mass on large sets, e.g., highest epistemic uncertainty is reflected by mass 1 on the full label set $\Theta$, we explicitly don’t want $h(a) \leq h(B)$ when $A \subseteq B$ in general. Note that the construction rules for belief and plausibility naturally ensure $bel(A) \leq bel(B)$ and $pl(A) \leq pl(B)$ for $A \subseteq B$.
>
> > Later on it stated that the approach "is applicable to arbitrary network architectures". I am afraid that I am entirely sold on this.
>
> As stated above, it is meant that it is applicable to arbitrary network architectures if the output layer is adjusted accordingly. We have incorporated this clarification in the updated version of the manuscript.
>
> Regarding 2.1:
> >The loss function in (8) involves the plausibility and belief outputs, which are computed from the mass (output of the basic classifier). To update the network, the gradient of the loss function has to be computed, which necessarily have to take the mass to the plausibility/belief computation into consideration. So it is not really a post-processing step, but rather a fairly integral step. Correct me if I am wrong.
>
> For the full ($2^n$-dimensional) classifier, one indeed needs to compute belief and plausibility values during training, as can be done in a subsequent processing step following the construction rules (eqs. (3) and (4)). We have changed the wording to make the type of this processing clearer.
>
> Note that the reduced approach uses Equation (9) as loss function, which no longer utilizes belief values (as they are no longer available in the reduced approach). Here, no such subsequent processing step is necessary, as the outputs of the network are directly interpreted as plausibility values.

---

> ### Author Response · Authors · 2023-11-20
> **Response 2/3**
>
> Regarding 2.2:
> > But later on the loss function is replaced as well. Do you mean that the basic classifier still has a 2^n dimensional output, but only n components are used in the loss function (the rest are discarded), or do you mean that basic classifier has an n dimensional output to begin with?
>
> We indeed mean that the n-dim. DS Classifier has an n-dimensional output to begin with, whose non-normalized outputs are interpreted as plausibilities of the singletons from DST. That is, instead of being able to predict the entire credal set as the full DS Classifier (Fig. 2), the reduced n-dim. DS Classifier only predicts upper bounds on the probabilities, resulting in a probability simplex that is larger than the credal set (cf. updated Fig. 8) but still a reasonable estimate of epistemic uncertainty in many cases. Note that this reduction is necessary to break the exponential scaling of output layer parameters of the full classifier and thus unlocks the benefits of our approach also for larger classification tasks.
>
> >If it is the latter case, then the difference between the two approaches are more substantial. Moreover, the second approach would not be related to the Dempster-Shafer Theory at all.
>
> We agree that the reduced approach no longer contains all information predictions of the full approach offer, but it still has a clear interpretation in DST. I.e., we interpret the outputs as plausibilities of the singletons from DST (mass of 1 on the full label set results in all singletons having a plausibility of 1). Predicting “larger sets” in that sense then reduces to predicting high plausibility on multiple singletons and “smaller sets” to only predicting only one singleton to have high plausibility.
> We extended Fig. 8 in Appendix C and associated discussions to further elucidate the interpretation of the reduced approach in DST.
>
> > Moreover, if the "post-processing step" is removed for the second approach, then an updated graphic representation is needed in addition to Figure 2, instead of the two approaches sharing the same figure. A dedicated figure may help clarify any confusion.
>
> See updated Fig. 8 in Appendix C.
>
> Regarding 3.:
> > More to the second approach: I do not quite understand the role of the parameter in loss function (9). Shouldn't both CE and MSE have a somewhat same/similar goal? If $\lambda = 0$, then wouldn't the second approach reduce to a typical classification method? In this case, the only novelty in the second approach would be a half-new loss function with the CP in the end.
>
> The main architectural difference between the proposed n-dimensional DS Classifier and standard ML classifiers is that the output in our approach is not normalized to one (e.g., by softmax activation) since we choose to apply a sigmoid activation function (all outputs are between 0 and 1, but their sum can be larger than 1). That is, the loss function for $\lambda = 0$ (CE) pushes more weight to all outcomes as it does not penalize to put more weight to the plausibility of the wrong outcome (they are multiplied by zero due to the one-hot encoding of the labels). Therefore, all outcomes that predict a plausibility of 1 in the correct position minimize the CE, which is also the case for the full-label set, i.e., a plausibility of 1 on all singletons. Increasing the $\lambda$-parameter in favor of the MSE will lead to a more focused / certain prediction with only high plausibility on one singleton set. We have updated the manuscript to achieve higher clarity on that by adding an explanation below eq. (9).

---

> ### Author Response · Authors · 2023-11-20
> **Response 3/3**
>
> Regarding 4.:
> >The procedure ends with the CP applied to the plausibility. Here the plausibility is used merely as a conformity score. One can't help wonder if the result is due to CP or due to the choice of the score. Can we achieve similar performance if CP is applied to the softmax score or any other score of a standard-dimensional classifier (n is the number of classes)? In reverse, an ablate study is needed to see how the methods perform without the CP method in the end.
>
> In our numerical experiments we compare both of our proposed methods to a Standard NN classifier equipped with CP on the softmax scores and show empirically that we obtain smaller sets on average. It is right that this scoring function, although widely used in literature, is not the only choice. However, it was shown in previous work that standard softmax classifiers are not able to reliable capture epistemic uncertainty [https://proceedings.mlr.press/v202/bengs23a.html] and therefore we would not expect that changing the scoring function of the same model yields smaller set sizes.
> Further, we included an additional appendix (Appendix F ) containing a study on toy data designed to show the impact of epistemic uncertainty on DS and softmax classifiers. We not only clarify the connection to the observed improvements concerning FN control, but also explicitly show outputs before conformal calibration so as to disentangle the contribution of the two methods.

---

### Official Review · Reviewer_HmNM · 2023-10-31

**Soundness:** 4 excellent
**Presentation:** 4 excellent
**Contribution:** 3 good
**Rating:** 8
**Confidence:** 2

**Summary:**

This paper leverages the Dempster-Shafer theory of evidence (DST) to build a probabilistic set predictor from any classifier architecture. A probabilistic set predictor is a model that assigns probabilities to all possible subsets of outcomes. Two new losses to train such neural networks are introduced. Those are based on the concepts of belief and plausibility from DST. The output of such a model is combined with conformal prediction to produce calibrated set predictions. It is empirically shown that sets constructed with this method are on average smaller than those constructed with a basic classifier and conformal prediction suggesting that probabilistic set predictors from DST are better at quantifying uncertainty than basic classifiers.

**Strengths:**

* The paper is very clearly written and easy to follow. In particular, the background allows a reader who is not familiar with the Demptser-Shaffer theory of evidence to easily get in.
* The method is novel to me but I have limited knowledge of related works.
* Experiments are convincing.
* Developing new methods for efficient uncertainty quantification is of high significance.
* The methodology is sound and I did not identify any flaws.

**Weaknesses:**

I didn't identify strong weaknesses in this paper.

A minor remark would be that in equation (2), there is a $\sum_{A \subseteq \Theta}$ and a $\forall A \subseteq \Theta$. Should the $\forall A \subseteq \Theta$ be removed? Also, should it be a sum over $2^\Theta$ ?

**Questions:**

I do not have any questions.

---

> ### Author Response · Authors · 2023-11-20
>
> We thank the reviewer for the time taken to review our work and for the positive feedback! We are glad that you found the principles of our paper clear and interesting and the contributions meaningful. The summary of our work that was given reflects that is was well understood.
>
> As the reviewer pointed out correctly, there was a notational error in eq. (2)  ($\forall A \subseteq \Theta$ was misplaced). We changed that in the updated version of our manuscript. We would like to emphasize that the sum over $A \subseteq \Theta$ is equivalent to the sum over all elements in $2^\Theta$.

---

### Official Review · Reviewer_oMQV · 2023-11-01

**Soundness:** 3 good
**Presentation:** 2 fair
**Contribution:** 2 fair
**Rating:** 5
**Confidence:** 3

**Summary:**

This paper presents a new approach to conformal prediction that makes use of non-conformity scores derived from the Demster-Shafer (DS) theory of evidence. Specifically, the authors train a network to minimize a loss based on matching DS-based plausibility and belief scores, that are assigned to all sets of possible outcomes (i.e., $2^{|\mathcal{Y}|}$ for classification). They also derive a more computationally friendly variation, that only assigns plausibility scores directly to $n$ singleton outcome sets. Empirically, the authors show that these scores can achieve smaller set sizes when plugged into a conformal prediction framework.

**Strengths:**

I found the discussion of the Dempster-Shafer theory of evidence interesting, and an appealing approach to disentangling aleatoric and epistemic uncertainty. It is also nice that it is relatively simple to implement for any low-cardinality classification problem (or higher cardinality with the authors' proposed simplification in Eq. (9)). The empirical results seem strong with respect to reducing set size, especially when exposed to perturbations at inference time. That said, I'm still not exactly clear as to what the true factors leading to its success are (see questions), and they are not compared to the strongest baselines.

**Weaknesses:**

I find the motivation of the paper hard to follow throughout, and lost the thread somewhat when it took a turn to considering CP and evaluating the reduction in set size vs. distinguishing epistemic from aleatoric uncertainty. While I liked the basic idea of Dempster-Shafer theory and its interpretation w.r.t. epistemic vs. aleatoric uncertainty, these advantages seem lost when only measuring set size. It seems that such an uncertainty framework is better used when epistemic vs. aleatoric uncertainty quantification is explicitly called for, such as in applications like active learning.

With respect to only measuring set size, this plausibility function simply reduces to another conformity measure, and it would be good to compare it to more competitive measures like RAPS, APS, conformalized bayesian outputs, or conformal methods such as jackknive+ that can adapt to changes in the calibration set by training.

Some other minor comments:
- In line citations are poorly formatted (should use \citep)
- The shadow fonts for p(A), p(B), P(C) are fairly strange (use normal font?)

**Questions:**

I'm a bit confused as to why the n-dim DS classifier handles epistemic uncertainty better than the softmax classifier, especially as demonstrated in Figure 3. As noted in the text, the n-dim classifier loses the ability to distinguish between aleatoric and epistemic uncertainty (since uncertainty is only able to be measured on the singletons, vs the larger sets). I understand that the softmax classifier would be at least a normalized version of this, but I'm not sure why it would completely switch its predictions in a way that assigns mass to a class completely ignored by the n-dim one (i.e., the {1} set).

This also seems intimately related to why it does worse in noised settings, as rather than being equally distributed between classes {0} and {2} (which would be the case if the logits of the n-dim classifer where simply softmax'd), it significantly reduces the mass on {0} in favor of {1} for some reason---and this should be what results in the $(1 - \alpha)$ quantile being poor. So I'm still not clear on why exactly this model "can be expected to frequently attribute high probability to incorrect labels", and the n-dim one is not (which will also lead to large set sizes if all labels have high scores).

---

> ### Author Response · Authors · 2023-11-20
> **Response 1/2**
>
> We thank the reviewer for their thoughtful analysis of our manuscript and for the suggestions on how to improve clarity about the advantages of our approach for the reader. In the following, we address the comments one by one, to facilitate easy tracking of raised questions / points and our answers.
>
> > As noted in the text, the n-dim classifier loses the ability to distinguish between aleatoric and epistemic uncertainty (since uncertainty is only able to be measured on the singletons, vs the larger sets)
>
> That is true and one of the reasons why we are not applying it to settings in which a distinction is necessary. As was pointed out by the reviewer earlier, the full DS classifier could be employed in those scenarios like active learning or more general making decisions sensitive to the type of uncertainty. Nevertheless, the n-dimensional classifier is able to naturally capture epistemic uncertainty, which already extends the capability of softmax classifiers and may be useful to some applications (such as FN control as demonstrated here).
>
> > I understand that the softmax classifier would be at least a normalized version of this, but I'm not sure why it would completely switch its predictions in a way that assigns mass to a class completely ignored by the n-dim one (i.e., the {1} set).
>
> We would like to point out that the n-dim. DS classifier differs to the softmax classifier not only in that it’s output is not normalized, but also substantially in the training process. The hyperparameter $\lambda$ in the loss function that we employ (eq. (9)) enables the ML practitioner to put more weight on “larger sets”, i.e., high plausibility on more than one singleton set (small $\lambda$) or more weight on “singleton sets”, i.e., high plausibility on only one singleton during the training phase. That is, because while the MSE enforces a one-hot encoded plausibility prediction, the CE pushes more weight to all outcomes as it does not penalize to put more weight to the plausibility of the wrong outcome (they are multiplied by zero due to the one-hot encoding). Therefore, all outcomes that predict a plausibility of 1 in the correct position minimize the CE, which is also the case for the full-label set, i.e., a plausibility of 1 on all singleton sets.
>
> > This also seems intimately related to why it does worse in noised settings, as rather than being equally distributed between classes {0} and {2} (which would be the case if the logits of the n-dim classifer where simply softmax'd), it significantly reduces the mass on {0} in favor of {1} for some reason---and this should be what results in the quantile being poor.
>
> The specific example in Fig. 3 has an illustrative purpose and is meant to emphasize that the DST-inspired loss function of the n-dim. DS classifier renders the model more robust under epistemic uncertainty as was empirically demonstrated by our experiments. We agree that this should be highlighted in the corresponding section and updated the manuscript correspondingly.
>
> > So I'm still not clear on why exactly this model "can be expected to frequently attribute high probability to incorrect labels", and the n-dim one is not (which will also lead to large set sizes if all labels have high scores).
>
> In the presence of epistemic uncertainty, the softmax model is incapable of representing the full uncertainty of the prediction. Hence, overconfident and wrong predictions will occur. We hope that emphasizing the importance and interpretation of the loss function of the n-dim. DS classifier for such settings has cleared up some of the doubts. As stated above, we adjusted the manuscript to clarify these considerations.

---

> ### Author Response · Authors · 2023-11-20
> **Response 2/2**
>
> >While I liked the basic idea of Dempster-Shafer theory and its interpretation w.r.t. epistemic vs. aleatoric uncertainty, these advantages seem lost when only measuring set size. It seems that such an uncertainty framework is better used when epistemic vs. aleatoric uncertainty quantification is explicitly called for, such as in applications like active learning. With respect to only measuring set size, this plausibility function simply reduces to another conformity measure, and it would be good to compare it to more competitive measures like RAPS, APS, conformalized bayesian outputs, or conformal methods such as jackknive+ that can adapt to changes in the calibration set by training.
>
> The aim of our work is to establish a new method to train classifiers using Dempster-Shafer theory as lightweight approach to capture epistemic (as well as aleatoric) uncertainty. As such, we refrain from further use case specific demonstrations of epistemic uncertainty utilization such as for active learning or corner case detection, but instead strengthen our paper by an additional appendix (Appendix F) illustrating  the capability of our approach to capture epistemic uncertainty in a controlled toy setting and the connection to set sizes obtained after conformal calibration.
>
> Needless to say, we make use of conformal prediction to demonstrate the usefulness of our epistemic uncertainty estimation for FNR control, but don’t develop a new calibration procedure in itself, which motivates the limited comparison with other literature calibration methods and focus on the impact / successful description of epistemic uncertainty in the added appendix.
>
> On a sidenote, the heuristically motivated regularization term of RAPS applies corrections on outcomes with smaller probability, as “noisy probability estimates [appear] far down the list of classes”. Interpreting this noisiness as epistemic uncertainty, our DS Classifier would assign high mass in the smaller sets of more certain classes and smaller masses to other sets, which are then combined to give low belief, but relatively (compared to the probability estimate) larger plausibility values for the singleton sets of less certain classes (see, e.g., Fig.3). It seems that both approaches would lead to similar corrections to the calibration scores (compared to probability estimates as calibration scores), albeit with different interpretations. While interesting, we think a more detailed analysis of parallels between these approaches or the role of epistemic uncertainty for conformal prediction in general is best left for future work / out of scope for this paper.

---

> > ### Comment · Reviewer_oMQV · 2023-11-22
> >
> > Thanks for your responses to my comments, and the corresponding revisions to the paper. I'm slightly increasing my score.

---

### Meta-Review · Area_Chair_1iRC · 2023-12-11

**Metareview:**

This work proposes classifiers based on the framework of Dempster-Shafer theory that can better deal with uncertainty quantification and still be efficient. There is a plenitude of literature on this and related topics. It was not clear to the committee how novel these ideas are, apart from the clever utilisation/combination with neural nets. This definitely has merits, but it fell short of convincing the majority of the committee members at this moment (perhaps the short reviewing period has hindered the result, but that is part of the system anyway).

**Justification For Why Not Higher Score:**

The reproducibility and impact are still unclear, and the connection to previous papers on similar topics was subpar. There are doubts about the amount/importance of the novelty because of that. The work seems to be well thought, though the presentation challenged experts.

**Justification For Why Not Lower Score:**

N/A

---

### Decision · Program_Chairs · 2024-01-16

Reject